# RobustSpring: Benchmarking Robustness to Image Corruptions for Optical Flow, Scene Flow and Stereo

**Victor Oei**[*][1]    **Jenny Schmalfuss**[* † 1,2]    **Lukas Mehl**[1]    **Madlen Bartsch**[1]
**Shashank Agnihotri**[3]    **Margret Keuper**[3,4]    **Andreas Bulling**[1]    **Andrés Bruhn**[1]

[1]University of Stuttgart,    [2]NVIDIA,    [3]University of Mannheim,    [4]MPI for Informatics

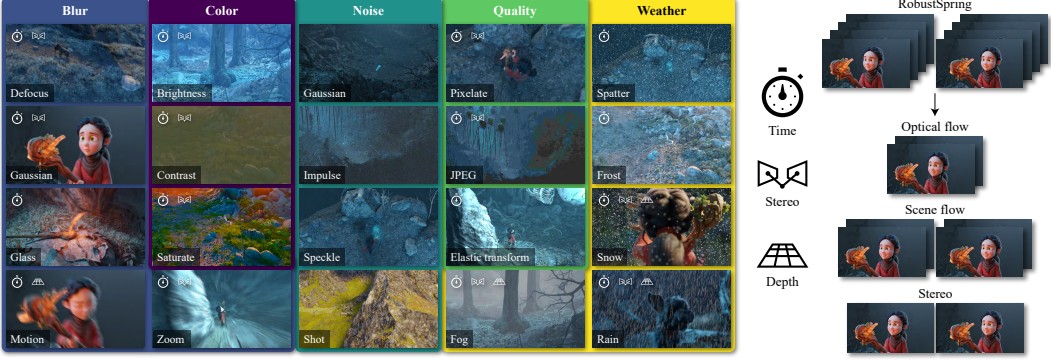

Figure 1: RobustSpring is a novel image corruption benchmark for optical flow, scene flow, and stereo. It evaluates 20 image corruptions including blurs, color changes, noises, quality degradations, and weather, applied to stereo video data from Spring. For comprehensive robustness evaluations, RobustSpring's image corruptions are integrated in time, stereo, and depth, where applicable.

## Abstract

Standard benchmarks for optical flow, scene flow, and stereo vision algorithms generally focus on model accuracy rather than robustness to image corruptions like noise or rain. Hence, the resilience of models to such real-world perturbations is largely unquantified. To address this, we present RobustSpring, a comprehensive dataset and benchmark for evaluating robustness to image corruptions for optical flow, scene flow, and stereo models. RobustSpring applies 20 different image corruptions, including noise, blur, color changes, quality degradations, and weather distortions, in a time-, stereo-, and depth-consistent manner to the high-resolution Spring dataset, creating a suite of 20,000 corrupted images that reflect challenging conditions. RobustSpring enables comparisons of model robustness via a new corruption robustness metric. Integration with the Spring benchmark enables two-axis evaluations of both accuracy and robustness. We benchmark a curated selection of initial models, observing that robustness varies widely by corruption type, and experimentally show that evaluations on RobustSpring indicate real-world robustness. RobustSpring is a new computer vision benchmark to treat robustness as a first-class citizen, fostering models that are accurate and resilient.

## 1 Introduction

Optical flow, scene flow, and stereo vision algorithms estimate dense correspondences and enable real-world applications like robot navigation (McGuire et al., 2017; Zhang et al., 2025; Lamberti et al., 2024), video processing (Mehl et al., 2024), structure-from-motion (Maurer et al., 2018; Phan et al., 2020), medical image registration (Mocanu et al., 2021), or surgical assistance (Rosa et al.,

---

[*]Equal contribution. Correspondence to `victor.oei@vis.uni-stuttgart.de`.

[†]Work done at the University of Stuttgart, Jenny Schmalfuss is currently affiliated with NVIDIA.

2019; Philipp et al., 2022). While estimation quality continuously improves on accuracy-driven benchmarks (Mehl et al., 2023b; Menze & Geiger, 2015; Butler et al., 2012; Baker et al., 2011; Scharstein et al., 2014; Geiger et al., 2012; Richter et al., 2017; Schöps et al., 2017), their robustness to real-world visual corruptions like noise or compression artifacts is rarely systematically assessed. This lack of systematic assessment is problematic, as better accuracy does not necessarily translate to improved robustness, and can even harm model robustness (Tsipras et al., 2019; Schmalfuss et al., 2022b). Though image data in KITTI (Menze & Geiger, 2015), Sintel (Butler et al., 2012), or Spring (Mehl et al., 2023b) comes with degradations like motion blurs, depth-of-field, or brightness changes, they result from real-world data capture or efforts to increase data realism, but were not included to systematically study model predictions under image corruptions. Broad corruption-robustness studies as they exist for image classification (Hendrycks & Dietterich, 2019; Müller et al., 2023), 3D object detection (Michaelis et al., 2019; Kong et al., 2023) or monocular depth estimation (Kar et al., 2022) are rare for dense-correspondence tasks, where studies are limited to specific degradations like weather (Schmalfuss et al., 2023) or low-light (Zheng et al., 2020). This not only leaves uncertainty about the reliability of dense matching algorithms in real-world scenarios. It also prevents systematic efforts to improve their robustness.

To enable systematic studies on the image corruption robustness of optical flow, scene flow, and stereo, we propose the *RobustSpring* dataset. Based on Spring (Mehl et al., 2023b), it jointly benchmarks robustness of all three tasks on corrupted stereo videos. While prior image corruptions affect the monocular 2D or 3D space (Hendrycks & Dietterich, 2019; Michaelis et al., 2019; Kar et al., 2022; Schmalfuss et al., 2025), RobustSpring's image corruptions are integrated in *time*, *stereo*, and *depth* and thus tailored to dense matching tasks. A principled corruption robustness metric and an accompanying benchmark framework make RobustSpring the first systematic tool to evaluate and improve dense matching robustness to image corruptions.

**Contributions.** Figure 1 gives an overview of RobustSpring. In summary, we make the following contributions:

(1) *Tailored image corruptions.* RobustSpring is the first image corruption dataset for optical flow, scene flow, and stereo. It integrates 20 corruptions for blurs, noises, tints, artifacts, and weather in time, stereo, and depth.

(2) *Corruption robustness metric.* We propose a corruption robustness metric based on Lipschitz continuity, which subsamples the clean-corrupted prediction difference and disentangles robustness and accuracy.

(3) *Benchmark functionality.* RobustSpring's standardized evaluation enables community-driven robustness comparisons of dense matching models. Public robustness benchmarking is integrated with Spring's website (https://spring-benchmark.org/).

(4) *Initial robustness evaluation.* We benchmark nine optical flow, two scene flow, and six stereo models. All models are corruption sensitive, which reveals concealed robustness deficits on dense matching models.

**Intended Use.** RobustSpring is not a fine-tuning dataset, but a benchmark of how dense matching models generalize to *unseen* image corruptions. It seeks to foster robustness research and, simultaneously, helps assess real-world applicability of models. Hence, it is essential to tie RobustSpring to an existing accuracy benchmark like Spring, as this minimizes the robustness evaluation hurdle for researchers. While RobustSpring treats corruptions as perturbations to assess robustness, their interpretation depends on the application domain. For instance, in autonomous driving, rain or snow are typically considered disturbances to be ignored for stable navigation, whereas in video editing or cinematic rendering, they may constitute meaningful scene content. To accommodate such differences, RobustSpring provides results per corruption type, allowing end-users to focus only on those corruptions that align with their intended application.

## 2 RELATED WORK

While the quality of optical flow, scene flow, and stereo models advanced for over three decades, their robustness recently regained attention as result of brittle deep learning generalization (Ranjan et al., 2019; Schmalfuss et al., 2022b). We review robustness in dense-matching, particularly image corruptions and metrics.

**Robustness in Dense Matching.** Robustness research for optical flow, scene flow, and stereo models often focuses on *adversarial attacks*, which quantify prediction errors for optimized image perturbations. Most attacks are for optical flow (Agnihotri et al., 2024c; Schmalfuss et al., 2023; 2022b; Schrodi et al., 2022; Ranjan et al., 2019; Yamanaka et al., 2021; Koren et al., 2022) rather than stereo (Berger et al., 2022; Wong et al., 2021) and scene flow (Wang et al., 2024a; Mahima et al., 2025). As remedies to adversarial vulnerability (Agnihotri et al., 2024b;a; 2023; Schrodi et al., 2022; Anand et al., 2020) may be overcome through specialized optimization (Scheurer et al., 2024), another line of robustness research considers non-adversarial data shifts. Those come in two flavors: *generalization across datasets*, *i.e.* the Robust Vision Challenge (http://www.robustvision.net/) or effective robustness (Bauer et al., 2025), and *robustness to image corruptions*. Work on dense matching models typically reports generalization (Mehl et al., 2023a; Teed & Deng, 2020; 2021; Lipson et al., 2021; Huang et al., 2022; Xu et al., 2022b) to several datasets, which span synthetic (Mehl et al., 2023b; Butler et al., 2012; Richter et al., 2017; Mayer et al., 2016; Dosovitskiy et al., 2015; Gaidon et al., 2016; Ranjan et al., 2020; Li et al., 2024) and real-world data (Geiger et al., 2012; Menze & Geiger, 2015; Kondermann et al., 2016; Scharstein et al., 2014; Schöps et al., 2017), often in automotive contexts. While some datasets contain image corruptions, *e.g.* motion blur, depth of field, fog, noise, or brightness changes (Sun et al., 2021; Butler et al., 2012; Mehl et al., 2023b; Menze & Geiger, 2015), they do not systematically assess corruption robustness. Yet, in the wild, robustness to image corruptions is crucial. For optical flow, systematic low light (Zheng et al., 2020) and weather datasets (Schmalfuss et al., 2022a; 2023) exist, and Schrodi et al. (2022); Yi et al. (2024) apply 2D image corruptions (Hendrycks & Dietterich, 2019) to optical flow data. Beyond these isolated works on optical flow, no systematic image-corruption study before RobustSpring spans all three dense matching tasks and includes scene flow or stereo.

**Robustness to Image Corruptions.** Popularized by 2D common corruptions (Hendrycks & Dietterich, 2019), the field of image corruption robustness rapidly expanded from classification (Hendrycks & Dietterich, 2019; Müller et al., 2023) to depth estimation (Kar et al., 2022), 3D object detection (Michaelis et al., 2019; Kong et al., 2023), and semantic segmentation (Kong et al., 2023). Conceptually, corruptions were extended to the 3D space (Kar et al., 2022), LiDAR (Kong et al., 2023), and procedural rendering (Drenkow & Unberath, 2024), but none have been tailored to the depth-, stereo-, and time-dependent setup of dense matching with optical flow, scene flow, and stereo.

**Robustness Metrics and Benchmarks.** Most robustness metrics for dense matching differ by whether they utilize ground truth (Ranjan et al., 2019; Agnihotri et al., 2024c; Yi et al., 2024) or not (Schmalfuss et al., 2022b; 2023; 2022a). However, multiple works (Schmalfuss et al., 2022b; Tsipras et al., 2019; Taori et al., 2020) show that robustness and accuracy are competing qualities that should not be quantified together. This informs our robustness metric. RobustSpring is the first dense-matching *robustness* benchmark, and joins prior classification robustness benchmarks (Croce et al., 2021; Jung et al., 2023; Tang et al., 2021)

# 3 ROBUSTSPRING DATASET AND BENCHMARK

RobustSpring is a large, novel image corruption dataset for optical flow, scene flow, and stereo. Below, we describe how we build on Spring's stereo video dataset and augment its frames with diverse image corruptions integrated in time, stereo, and depth, how we evaluate robustness to image corruptions, and use it to benchmark algorithm capabilities.

**Spring Data.** Spring (Mehl et al., 2023b) is a high-resolution benchmark with rendered stereo sequences and dense ground truth. It is the ideal base for an image corruption dataset as its detailed renderings permit image alterations of varying granularity – from removing detail by blurring to adding detail via weather. Spring provides a public training and closed test split, where test ground truth for optical flow, disparity, and extrinsic camera parameters is withheld. As RobustSpring is designed to complement accuracy analyses, we build on the 2000 Spring test frames (two per stereo camera). To apply corruptions with time, stereo, and depth consistency, we require depth and extrinsics that are not publicly available. We therefore estimate extrinsics using COLMAP 3.8 and depths via $Z = \frac{f_x \cdot B}{d}$, with focal length $f_x$, baseline $B$, and disparities $d$ predicted by MS-RAFT+ (Jahedi et al., 2022; 2024). This estimation avoids ground-truth leakage while maintaining benchmark integrity. Quantitative results on the accuracy of our depth and extrinsics estimation are given in App. A.5, and a detailed discussion of motion ranges in Spring is provided in App. A.7.

(a) Image corruptions on a single image.

| Property | Color | | | Blur | | | | | Noise | | | | Qual | | | Weather | | | | |
|---|---|---|---|---|---|---|---|---|---|---|---|---|---|---|---|---|---|---|---|---|
| | Brightness | Contrast | Saturate | Defocus | Gaussian | Glass | Motion | Zoom | Gaussian | Impulse | Speckle | Shot | Pixelate | JPEG | Elastic | Spatter | Frost | Snow | Rain | Fog |
| Time-cons. | ✓ | ✓ | ✓ | ✓ | ✓ | ✓ | ✓ | ✓ | – | – | – | – | ✓ | ✓ | ✓ | ✓ | ✓ | ✓ | ✓ | ✓ |
| Stereo-cons. | ✓ | ✓ | ✓ | ✓ | ✓ | – | – | ✓ | – | – | – | – | ✓ | ✓ | – | – | – | ✓ | ✓ | ✓ |
| Depth-cons. | – | – | – | – | – | – | ✓ | – | – | – | – | – | – | – | – | – | – | ✓ | ✓ | ✓ |
| SSIM | 0.70 | 0.70 | 0.72 | 0.70 | 0.70 | 0.73 | 0.75 | 0.70 | 0.20 | 0.20 | 0.20 | 0.22 | 0.70 | 0.70 | 0.70 | 0.72 | 0.73 | 0.70 | 0.70 | 0.71 |

(b) Overview of corruptions and their consistency in time, stereo, or depth, with resulting visual changes w.r.t. the original images as SSIM.

Figure 2: Overview of RobustSpring's image corruptions.

### 3.1 CORRUPTION DATASET CREATION

RobustSpring corrupts the Spring test frames via 20 diverse image corruptions, summarized in Fig. 2a and Fig. 2b. Below, we describe the image corruption types, their new consistencies, their implementation, and their severity levels.

**Corruption Types.** In RobustSpring, we consider the five image corruption types from Hendrycks & Dietterich (2019): color, blur, noise, quality, and weather. Color simulates different lighting conditions and camera settings, including brightness, contrast, and saturation. Blur acts like focus and motion artifacts, including defocus, Gaussian, glass, motion, and zoom blur. Noise represents sensor errors and ambiance, including Gaussian, impulse, speckle, and shot noise. Quality distortions are lossy compressions and geometric distortions, including pixelation, JPEG, and elastic transformations. Weather enacts outdoor conditions, including spatter, frost, snow, rain, and fog. All corruptions applied to the same frame are shown in Fig. 2a.

While these 20 corruptions do not cover the entire corruption space, they are chosen to represent the most common perturbations encountered in real imagery and to provide a balanced basis for robustness evaluation. Several implementations were adapted and we additionally include practically relevant corruptions such as saturation, Gaussian blur, speckle noise, rain, and spatter. The blur and noise families span the most dominant optical and sensor degradations, and the weather and quality corruptions capture major outdoor and compression-related effects. At the same time, some perturbations, such as illumination changes requiring re-rendering (*e.g.* colored or dynamic lighting), lie beyond what can be approximated in post-processing. These, along with further outdoor effects (*e.g.* bloom, glare, dusty conditions) or extended codec distortions (*e.g.* JPEG 2000), form natural directions for future extensions.

**Corruption Consistencies.** To increase the realism of these 20 corruptions for dense matching models, we extend their definition to time, stereo, and depth: *Time-consistent* corruptions evolve smoothly over subsequent frames for a single camera, mirroring persistent lens or sensor effect, *e.g.* frost follows a temporally coherent pattern on one camera but differs between left and right. *Stereo-consistent* corruptions equally influence both stereo cameras, such as shared brightness or contrast adjustments. Unlike simply using the same hyperparameters, stereo consistency does not imply identical pixel-wise noise realizations, only that both views undergo the same transformation strength. *Depth-consistent* corruptions are rendered directly in the 3D scene, ensuring that their projection into each stereo view respects geometry. This applies to weather effects, such as snow, rain, and fog, where particles follow 3D trajectories and generate view-dependent projections. Other corruptions, such as blur or noise, do not benefit from 3D rendering; therefore, they use independent realizations per frame, despite sharing global severity parameters. Fig. 2b summarizes the consistencies we added to 16 of our 20 corruptions. Note that motion blur is not stereo-consistent because it depends on the specific camera view.

**Corruption Implementation.** Though most corruptions are loosely based on Hendrycks & Dietterich (2019), our corruption consistencies require multiple adaptations. Furthermore, we employ specialized techniques for highly consistent effects, *i.e.* motion blur, elastic transform, snow, rain and fog. We adapt implementations from Hendrycks & Dietterich (2019), modify glass blur, zoom blur, frost, and pixelation to accommodate higher resolutions and non-square images, and adjust

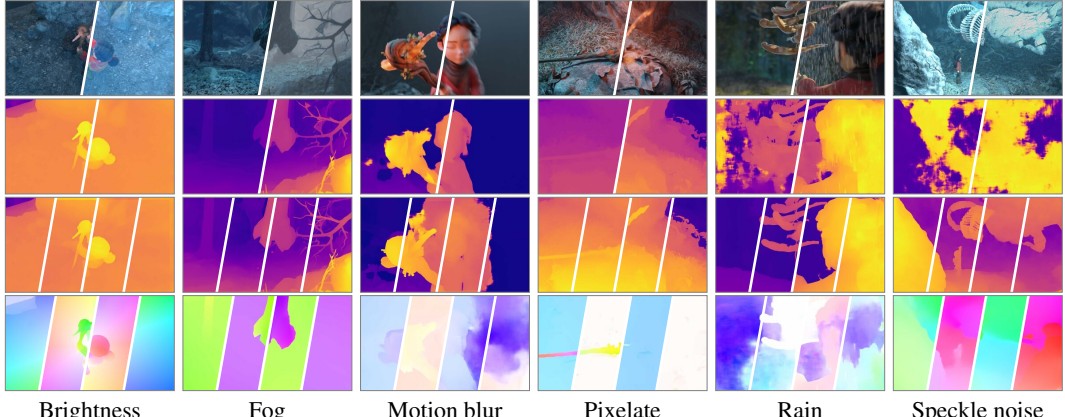

Figure 3: RobustSpring example frames. First row shows clean and corrupted images. Second row shows the left and right disparity maps predicted with LEA Stereo (Cheng et al., 2020). Third row shows the target disparities for forward left, backward left, forward right, and backward right directions from M-FUSE (Mehl et al., 2023a). Fourth row shows optical flow estimates for forward left, backward left, forward right, and backward right from RAFT (Teed & Deng, 2020). All disparities and flows are computed on the corrupted dataset. See Fig. 6 in App. A.3.1 for additional frames.

frost, glass blur, and spatter for consistency across video scenes. Motion blur is based on Zheng et al. (2006) and adds camera-induced motion with clean optical flow estimates. Elastic transform uses PyTorch's transforms package to create a see-through-water-like effect, changing object morphology with smooth frame transitions. For snow and rain, we expand the two-step 3D particle rendering of Schmalfuss et al. (2023) to multi-step particle trajectories and stereo views, change from additive-blending to order-independent alpha blending (McGuire & Bavoil, 2013), and include global illumination (Halder et al., 2019). To augment the large-scale Spring data, we improve its performance via more effective particle generation and parallel processing. Fog is based on the Koschmieder model following Wiesemann & Jiang (2016). Full details are in App. A.3.3.

**Corruption Severity.** Prior works (Hendrycks & Dietterich, 2019; Müller et al., 2023; Kar et al., 2022; Michaelis et al., 2019; Kong et al., 2023) defined corruptions with several levels of severity. Here, we opt for one severity per corruption, because evaluating one scene flow model on all 20 corruptions already produces 2.1 TB of raw data – 1.2 GB after subsampling, *cf.* Sec. 3.2. More severity levels would overburden the evaluation resources of benchmark users. To balance severity across corruptions, we tune their hyperparameters until the image SSIM reaches a defined threshold. We generally use SSIM $\geq 0.7$, and, because the SSIM is less sensitive to blurs than noises (Hore & Ziou, 2010), SSIM $\geq 0.2$ for noises for visually similar artifact strengths. We conducted a focused perceptual study to validate the selection of corruption strengths and their corresponding SSIM values in RobustSpring. See App. A.6 for more details. Final SSIMs are in Fig. 2b.

## 3.2 ROBUSTNESS EVALUATION METRIC

With various corruption types, we need a metric to quantify model robustness to these variations. In the following, we motivate and derive a ground-truth-free robustness metric for dense matching, introduce subsampling for efficiency, and discuss strategies for joint rankings over corruptions.

**Definition of Optical Flow.** Throughout this work, and consistent with the Spring benchmark (Mehl et al., 2023b), we define optical flow as the *true 3D motion of visible surfaces projected into the 2D image plane* (Horn & Schunck, 1981; Baker et al., 2011). This definition is standard in classical benchmarks and provides the basis for accuracy evaluation. Alternative formulations, such as apparent motion, exist in the literature, but RobustSpring's robustness evaluation is agnostic to this choice, since robustness is measured independently of ground truth. A more detailed discussion on the implications of this definition, including the relation between perfect accuracy and perfect robustness, is provided in App. A.1.

**Robustness Metric Concepts.** For dense matching, robustness to corruptions lacks a standardized evaluation metric. Metrics exist for adversarial robustness, using the distance between corrupt

prediction and either (i) ground-truth (Ranjan et al., 2019; Agnihotri et al., 2024c) or (ii) clean prediction (Schmalfuss et al., 2022b; 2023; 2022a). The latter is preferred for two reasons: First, (i)'s ground-truth comparisons mix accuracy and robustness, which are competing model qualities (Schmalfuss et al., 2022b; Tsipras et al., 2019; Taori et al., 2020) that should be separate. This competition is intuitive: A model that always outputs the same value is as robust as inaccurate. Likewise, an accurate model varies for any input change and thus is not robust. Second, (ii) separates robustness from accuracy and builds on an established mathematical concept for system robustness (Hein & Andriushchenko, 2017; Pauli et al., 2022): the Lipschitz constant $L^c$. It defines robust models as those whose prediction $f$ is similar on clean and corrupt image $I$ and $I^c$, relative to their difference. For dense matching, it reads

$$L^c = \frac{\|f(I) - f(I^c)\|}{\|I - I^c\|},\tag{1}$$

where the term $\|I - I^c\|$ refers to the per-pixel intensity difference between the clean and corrupted images. This robustness formulation is preferable for real-world applications that demand stable scene estimations *despite* corruptions like snow, and remains valid independent of whether optical flow is defined as true motion or apparent motion. We emphasize that RobustSpring explicitly measures robustness in terms of *stability*: models are considered robust if their predictions remain consistent under corrupted inputs. Lower robustness scores correspond to higher stability, not improved accuracy. Other definitions of robustness, such as ground-truth-based robustness, are possible and may be more appropriate for different use cases. Our benchmark deliberately focuses on stability-based robustness, providing a complementary axis of evaluation alongside accuracy.

**Corruption Robustness Metric.** Based on Eq. (1), we quantify model robustness to corruptions. Because RobustSpring's corrupt images $I_c$ deviate from their clean counterparts $I$ by a similar amount, *cf.* SSIM equalization in Sec. 3.1, we omit the denominator in Eq. (1) and define *corruption robustness* $R^c$ as distance between clean $f(I)$ and corrupted $f(I^c)$ predictions with distance metric M:

$$R^c_{\mathrm{M}} = \mathrm{M}[f(I), f(I^c)].\tag{2}$$

For similarity to Spring's evaluation, we use corruption robustness with various metrics M, reporting $R^c_{\mathrm{EPE}}$, $R^c_{\mathrm{1px}}$ and $R^c_{\mathrm{Fl}}$ for optical and scene flow, and $R^c_{\mathrm{1px}}$, $R^c_{\mathrm{Abs}}$ and $R^c_{\mathrm{D1}}$ for stereo. Here, EPE denotes the average end-point error, 1px the percentage of pixels with an error exceeding 1 px, Fl the KITTI optical-flow outlier rate, Abs the mean absolute disparity error, and D1 the KITTI disparity outlier rate, see Mehl et al. (2023b) and Menze & Geiger (2015) for more details. Interestingly, our EPE-based corruption robustness

$$R^c_{\mathrm{EPE}} = \mathrm{EPE}[f(I), f(I^c)] = \frac{1}{|\Omega|}\sum_{i\in\Omega}\|f_i(I) - f_i(I^c)\|\tag{3}$$

on image domain $\Omega$ is a generalization of optical-flow adversarial robustness (Schmalfuss et al., 2022b) to dense matching and corruptions.

**Metric Subsampling.** For a benchmark, users should upload robustness results to a web server. Given the large number of 20 datasets, data reduction is essential to facilitate evaluations and uploads. To this end, we evaluate on a reduced set of pixels by refining the original subsampling strategy from Spring, which retains about 1% of the full data. First, we additionally subsample the set of full-resolution Hero-frames because they are only kept in the original benchmark for visualization purposes, leaving 0.95%, and then apply 20-fold subsampling, ultimately keeping 0.05% of the full data.

**Robustness Ranking.** Because we generate 20 different corruption evaluations *per* dense matching model, we need a summarization strategy to produce one result per model. Per-model results are ranked based on three strategies: Average, Median, and the Schulze voting method (Schulze, 2018). In contrast to averaging across all 20 evaluations, the median reduces the impact of extreme outliers. The Schulze method provides a holistic, pairwise comparison approach that ranks models based on preference aggregation and was used for prior generalization evaluations in the Robust Vision Challenges. We evaluate their differences in Sec. 4.2.

### 3.3 DATASET AND BENCHMARK FUNCTIONALITY

Below, we summarize RobustSpring's corruption dataset and describe its benchmark function. Fig. 3 shows data samples with stereo, optical flow, and scene flow estimates.

Table 1: Initial RobustSpring results on corruption robustness of optical flow models, using $R_{\text{EPE}}^c$, $R_{\text{1px}}^c$ and $R_{\text{Fl}}^c$ between clean and corrupted flow predictions. Low values indicate robust models. $\varepsilon_{\text{clean}}$ compares clean predictions with ground-truth flow, values from Mehl et al. (2023b).

| | | SEA-RAFT | | | GMFlow | | | MS-RAFT+ | | | FlowFormer | | | GMA | | | SPyNet | | | RAFT | | | FlowNet2 | | | PWCNet | | |
|---|---|---|---|---|---|---|---|---|---|---|---|---|---|---|---|---|---|---|---|---|---|---|---|---|---|---|---|---|
| | | $R_{\text{EPE}}^c$ | $R_{\text{1px}}^c$ | $R_{\text{Fl}}^c$ | $R_{\text{EPE}}^c$ | $R_{\text{1px}}^c$ | $R_{\text{Fl}}^c$ | $R_{\text{EPE}}^c$ | $R_{\text{1px}}^c$ | $R_{\text{Fl}}^c$ | $R_{\text{EPE}}^c$ | $R_{\text{1px}}^c$ | $R_{\text{Fl}}^c$ | $R_{\text{EPE}}^c$ | $R_{\text{1px}}^c$ | $R_{\text{Fl}}^c$ | $R_{\text{EPE}}^c$ | $R_{\text{1px}}^c$ | $R_{\text{Fl}}^c$ | $R_{\text{EPE}}^c$ | $R_{\text{1px}}^c$ | $R_{\text{Fl}}^c$ | $R_{\text{EPE}}^c$ | $R_{\text{1px}}^c$ | $R_{\text{Fl}}^c$ | $R_{\text{EPE}}^c$ | $R_{\text{1px}}^c$ | $R_{\text{Fl}}^c$ |
| Color | Brightness | 0.21 | 1.65 | 0.40 | 0.33 | 3.31 | 1.12 | 0.33 | 2.88 | 1.02 | 0.68 | 2.82 | 1.05 | 0.36 | 3.22 | 1.04 | 2.72 | 14.67 | 8.91 | 0.92 | 3.49 | 1.61 | 0.45 | 3.16 | 1.05 | 1.04 | 7.38 | 3.00 |
| | Contrast | 0.75 | 3.75 | 1.51 | 0.46 | 6.71 | 1.71 | 0.87 | 6.69 | 3.24 | 0.93 | 5.48 | 1.96 | 0.68 | 6.43 | 2.20 | 8.23 | 38.90 | 27.23 | 1.32 | 5.73 | 2.64 | 1.87 | 9.26 | 4.74 | 2.98 | 30.07 | 7.42 |
| | Saturate | 0.16 | 1.29 | 0.36 | 0.34 | 3.30 | 0.96 | 0.34 | 2.87 | 1.03 | 0.42 | 2.39 | 0.88 | 0.43 | 3.47 | 1.18 | 3.36 | 17.34 | 11.31 | 0.93 | 3.33 | 1.47 | 0.51 | 3.40 | 1.10 | 1.21 | 9.92 | 3.68 |
| Blur | Defocus | 0.20 | 1.47 | 0.42 | 0.53 | 6.17 | 1.45 | 0.51 | 4.01 | 1.47 | 0.55 | 3.85 | 1.19 | 0.56 | 5.02 | 2.01 | 0.57 | 10.16 | 1.36 | 1.03 | 4.70 | 2.07 | 0.53 | 3.35 | 1.06 | 0.98 | 6.51 | 2.78 |
| | Gaussian | 0.23 | 1.79 | 0.51 | 0.66 | 7.77 | 1.88 | 0.58 | 4.45 | 1.63 | 0.63 | 4.32 | 1.37 | 0.62 | 5.48 | 2.22 | 0.76 | 15.44 | 2.12 | 1.10 | 5.12 | 2.26 | 0.60 | 4.05 | 1.27 | 1.11 | 7.72 | 3.09 |
| | Glass | 0.22 | 1.46 | 0.41 | 0.85 | 20.87 | 1.82 | 0.53 | 4.45 | 1.37 | 0.64 | 4.04 | 1.17 | 0.61 | 5.60 | 1.91 | 0.75 | 16.94 | 1.36 | 1.05 | 5.13 | 1.97 | 0.50 | 3.12 | 0.96 | 0.91 | 5.96 | 2.47 |
| | Motion | 1.11 | 13.78 | 5.82 | 1.34 | 18.35 | 7.51 | 1.31 | 14.06 | 6.16 | 1.35 | 14.03 | 5.77 | 1.19 | 14.40 | 6.18 | 2.32 | 19.55 | 10.05 | 2.06 | 14.33 | 6.35 | 1.60 | 14.07 | 6.47 | 1.95 | 16.25 | 7.47 |
| | Zoom | 1.83 | 24.51 | 8.41 | 1.88 | 35.80 | 9.90 | 1.81 | 21.84 | 7.13 | 1.66 | 22.72 | 6.77 | 1.54 | 23.17 | 7.16 | 4.82 | 46.67 | 28.37 | 3.14 | 22.80 | 7.61 | 2.36 | 24.63 | 9.04 | 3.52 | 50.33 | 15.64 |
| Noise | Gaussian | 1.49 | 12.30 | 4.34 | 4.70 | 57.95 | 21.67 | 5.70 | 35.74 | 22.12 | 6.56 | 27.83 | 18.30 | 2.81 | 24.70 | 12.96 | 2.22 | 42.23 | 14.88 | 7.43 | 27.92 | 18.99 | 1.33 | 11.24 | 5.06 | 2.79 | 26.87 | 9.89 |
| | Impulse | 2.58 | 19.01 | 7.51 | 6.64 | 66.14 | 28.70 | 7.39 | 45.72 | 29.05 | 7.33 | 23.58 | 14.47 | 4.08 | 31.31 | 18.13 | 2.92 | 53.45 | 20.41 | 6.51 | 29.65 | 18.32 | 2.37 | 15.70 | 7.48 | 3.57 | 35.67 | 14.45 |
| | Speckle | 1.29 | 12.15 | 3.61 | 3.90 | 62.01 | 20.64 | 4.22 | 34.96 | 17.18 | 5.47 | 25.52 | 15.60 | 5.32 | 25.22 | 12.66 | 1.95 | 46.32 | 12.89 | 6.62 | 26.05 | 16.48 | 1.32 | 12.57 | 4.19 | 2.74 | 26.83 | 8.00 |
| | Shot | 1.20 | 10.69 | 3.41 | 3.52 | 56.71 | 17.77 | 4.36 | 31.67 | 17.77 | 5.75 | 26.02 | 16.01 | 3.15 | 23.11 | 11.59 | 1.86 | 40.44 | 11.98 | 6.74 | 25.64 | 17.08 | 1.16 | 9.87 | 3.92 | 2.59 | 23.75 | 7.88 |
| Quality | Pixelate | 0.41 | 1.88 | 0.51 | 1.96 | 68.09 | 18.71 | 1.60 | 45.83 | 6.78 | 1.48 | 31.68 | 2.59 | 1.11 | 25.86 | 1.78 | 1.22 | 50.63 | 2.90 | 1.65 | 21.47 | 2.00 | 0.77 | 7.74 | 0.88 | 0.92 | 8.67 | 2.22 |
| | JPEG | 4.02 | 34.86 | 13.57 | 3.32 | 83.54 | 27.92 | 2.09 | 41.69 | 12.82 | 2.89 | 42.62 | 14.96 | 1.92 | 38.70 | 11.51 | 2.95 | 53.97 | 18.08 | 3.19 | 37.72 | 13.67 | 2.56 | 31.00 | 11.85 | 2.88 | 49.15 | 15.91 |
| | Elastic | 0.56 | 11.14 | 1.42 | 1.37 | 40.00 | 6.89 | 1.16 | 32.49 | 5.54 | 2.62 | 35.78 | 11.01 | 1.24 | 27.24 | 6.40 | 1.08 | 34.62 | 4.77 | 1.33 | 19.43 | 4.78 | 0.79 | 16.27 | 2.12 | 1.42 | 28.18 | 5.47 |
| Weather | Fog | 1.20 | 11.20 | 7.41 | 0.80 | 14.42 | 5.32 | 0.91 | 10.32 | 6.33 | 0.86 | 9.66 | 5.67 | 0.84 | 11.21 | 6.42 | 5.20 | 28.15 | 19.97 | 1.97 | 12.01 | 7.11 | 1.74 | 11.77 | 7.82 | 16.84 | 20.96 | 12.89 |
| | Frost | 7.60 | 36.86 | 21.86 | 8.20 | 63.96 | 29.96 | 7.38 | 29.96 | 21.25 | 8.18 | 34.19 | 23.87 | 8.13 | 34.30 | 22.31 | 6.97 | 45.13 | 30.13 | 8.37 | 32.75 | 21.76 | 7.22 | 33.69 | 21.15 | 8.27 | 50.31 | 27.44 |
| | Rain | 16.73 | 36.87 | 25.09 | 8.60 | 64.20 | 32.72 | 19.99 | 36.74 | 31.22 | 11.13 | 33.50 | 20.83 | 33.00 | 43.98 | 36.18 | 18.20 | 68.87 | 56.38 | 42.41 | 38.89 | 31.99 | 63.71 | 48.25 | 41.15 | 40.18 | 73.51 | 57.05 |
| | Snow | 8.54 | 60.52 | 43.81 | 3.60 | 70.60 | 29.90 | 4.69 | 33.21 | 30.91 | 7.92 | 40.20 | 33.82 | 5.30 | 40.82 | 33.35 | 12.08 | 74.27 | 66.65 | 7.16 | 37.04 | 31.37 | 39.79 | 68.67 | 61.60 | 39.73 | 90.80 | 81.91 |
| | Spatter | 8.93 | 53.31 | 30.52 | 6.58 | 67.90 | 27.09 | 6.63 | 28.22 | 20.24 | 8.41 | 40.38 | 26.92 | 7.75 | 36.11 | 21.81 | 5.71 | 48.60 | 33.82 | 7.98 | 30.37 | 19.87 | 9.13 | 45.03 | 28.99 | 9.33 | 65.41 | 40.19 |
| **Average** | | 2.96 | 17.52 | 9.05 | 2.98 | 40.89 | 14.68 | 3.62 | 23.39 | 12.21 | 3.77 | 21.53 | 11.21 | 4.03 | 21.47 | 10.95 | 4.29 | 38.32 | 19.18 | 5.64 | 20.18 | 11.47 | 7.01 | 18.84 | 11.09 | 7.25 | 31.71 | 16.44 |
| Std. Dev. | | 4.29 | 17.98 | 12.08 | 2.70 | 27.91 | 11.91 | 4.58 | 15.54 | 10.62 | 3.44 | 14.37 | 9.94 | 7.23 | 13.67 | 10.55 | 4.38 | 18.35 | 17.60 | 9.10 | 12.55 | 9.98 | 15.94 | 17.93 | 15.87 | 11.83 | 24.43 | 20.79 |
| **Median** | | 1.20 | 11.68 | 3.98 | 1.92 | 48.35 | 13.83 | 1.71 | 29.09 | 6.95 | 2.14 | 24.55 | 8.89 | 1.39 | 23.93 | 6.79 | 2.82 | 41.33 | 13.88 | 2.60 | 22.13 | 7.36 | 1.47 | 12.17 | 4.90 | 2.77 | 26.85 | 7.94 |
| $\varepsilon_{\text{clean}}$ (Clean Error) | | 0.36 | 3.69 | 1.35 | 0.94 | 10.36 | 2.95 | 0.64 | 5.72 | 2.19 | 0.72 | 6.51 | 2.38 | 0.91 | 7.07 | 3.08 | 4.16 | 29.96 | 12.87 | 1.48 | 6.79 | 3.20 | 1.04 | 6.71 | 2.82 | 2.29 | 82.27 | 4.89 |

**RobustSpring Dataset.** The final RobustSpring dataset entails 20 corrupted versions of Spring, resulting in 40,000 frames, or 20,000 stereo frame pairs. Each corruption evaluation yields 3960 optical flows (990 per camera & direction), 2000 stereo disparities (1000 per camera), and 3960 additional scene flow disparity maps (990 per camera per direction). To discourage corruption fine-tuning and ensure a fair benchmark, we do not provide corrupted training data, only corrupted test data. We separately provide the raw data and a curated dataset for predicting dense matches.

**RobustSpring Benchmark.** RobustSpring enables uploading robustness results to a benchmark website for display in a public ranking. To emphasize that robustness and accuracy are two axes of model performance with equal importance (Tsipras et al., 2019), we couple RobustSpring with Spring's established accuracy benchmark. Thus, researchers can report model robustness and accuracy on the same dataset. We provide mock-ups of this integration in App. A.8.

## 4 RESULTS

We evaluate RobustSpring under two aspects: First, we report initial results for 17 optical flow, scene flow and stereo models. Then, we analyze the benchmark evaluation, particularly subsampling strategy and ranking methods.

### 4.1 INITIAL ROBUSTSPRING BENCHMARK RESULTS

We provide initial results on RobustSpring for selected models from all three dense matching tasks. For optical flow, we include SEA-RAFT (Wang et al., 2024b), GMFlow (Xu et al., 2022b), MS-RAFT+ (Jahedi et al., 2024), FlowFormer (Huang et al., 2022), GMA (Jiang et al., 2021), SPyNet (Ranjan & Black, 2017), RAFT (Teed & Deng, 2020), FlowNet2 (Ilg et al., 2017), and PWCNet (Sun et al., 2018). For scene flow, we evaluate M-FUSE (Mehl et al., 2023a) and RAFT-3D (Teed & Deng, 2021). For stereo estimation, we evaluate RAFT-Stereo (Lipson et al., 2021), ACVNet (Xu et al., 2022a), LEAStereo (Cheng et al., 2020), and GANet (Zhang et al., 2019). An overview of all models and used checkpoints is in the Appendix in Tab. 4. Importantly, none of these models are fine-tuned to either Spring or RobustSpring data, to assess the generalization capacity of existing models.

**Optical Flow.** The evaluation results in Tab. 1 and Fig. 4a show large robustness variations across corruption types. Weather corruptions, especially rain and snow, degrade performance most, while color corruptions have little effect. Model rankings also vary: FlowNet2 performs poorly overall but is the most resilient to noise (Fig. 4b). SEA-RAFT and GMFlow achieve the lowest average $R_{\text{EPE}}^c$, while GMA yields the lowest median, as detailed in Sec. 4.2.

To examine accuracy–robustness relations, we compare both in Fig. 4c. Accurate models tend to be more robust, though no method excels in both dimensions, and the trend is weak and nonlinear (see

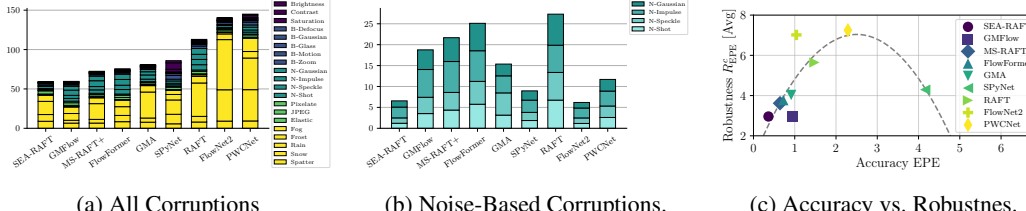

(a) All Corruptions      (b) Noise-Based Corruptions.      (c) Accuracy vs. Robustnes.

Figure 4: Accumulated corruption robustness $R^c_{\text{EPE}}$ for optical flow models over all corruptions *[left]*, only noise corruptions *[middle]*, and accuracy vs. robustness *[right]* with a dashed line representing a quadratic polynomial fit. Small values are robust (and accurate) models. All other corruption classes color (purple), blur (blue), noise (cyan), quality (green), and weather (yellow) are in App. A.4 in Fig. 7, and Fig. 8 shows accuracy vs. Median $R^c_{\text{EPE}}$.

also median robustness in Fig. 8). Fitting a quadratic regression to Fig. 4c highlights this nonlinearity. Accuracy and robustness coincide for some corruption families but diverge substantially for others. This stands in contrast to adversarial robustness, where a strong accuracy–robustness trade-off is observed (Schmalfuss et al., 2022b). The key to understanding these differences is to consider different corruption types individually. Because accurate models need to process fine details in order to achieve highly accurate predictions, this attention to details can be exploited by noise-based adversarial attacks in Schmalfuss et al. (2022b). For RobustSpring's non-adversarial corruptions, accurate models are also relatively sensitive to noise corruptions (*cf.* Fig. 4b). However, the average corruption robustness is mostly dominated by weather corruptions (*cf.* Fig. 4a), where more accurate models achieve better robustness values. We identify the reason for this in Sec. 4.2: accurate models retain corruption-induced motion errors around the particles rather than spreading them onto the background (*cf.* Fig. 5a). This aligns with the results in Schmalfuss et al. (2023), which found accurate models to be more robust towards adversarial weather attacks.

Architectural trends also emerge. Transformer models (GMFlow, FlowFormer) perform best overall but struggle with noise, likely due to global matching. Hierarchical models (*e.g.* MS-RAFT+) show balanced robustness and may benefit from multi-scale feature processing to cope with quality degradations. Stacked models (*e.g.* SEA-RAFT, FlowNet2) uniquely resist noise, possibly due to their progressive refinement across layers. Overall, architecture influences robustness to specific corruptions, but no paradigm is universally superior.

**Scene Flow.** The results for scene flow are in Tab. 2a, and include optical flow and target frame disparity predictions for M-FUSE and RAFT-3D. M-FUSE generally produces more robust optical flow across corruptions with a lower average $R^c_{\text{EPE}}$ than RAFT-3D. But both methods suffer sig-

Table 2: Initial RobustSpring results on corruption robustness of scene flow and stereo disparity models, using corruption robustness $R^c_{\text{1px}}$, $R^c_{\text{Abs}}$ and $R^c_{\text{Dl}}$ between clean and corrupted predictions. Low values indicate robust models. Corresponding Disparity 1 from scene flow models LEAStereo (s) for M-FUSE, and GANet (s) for RAFT-3D in Tab. 2b. Stereo disparity models use Stereo (s) and KITTI (k) checkpoints.

(a) Initial scene flow evaluation.

| | | M-FUSE | | | | | | RAFT-3D | | | | | |
| | | Optical flow | | | Disparity 2 | | | Optical flow | | | Disparity 2 | | |
| | | $R^c_{\text{EPE}}$ | $R^c_{\text{1px}}$ | $R^c_{\text{Fl}}$ | $R^c_{\text{Abs}}$ | $R^c_{\text{1px}}$ | $R^c_{\text{D2}}$ | $R^c_{\text{EPE}}$ | $R^c_{\text{1px}}$ | $R^c_{\text{Fl}}$ | $R^c_{\text{Abs}}$ | $R^c_{\text{1px}}$ | $R^c_{\text{D2}}$ |
|---|---|---|---|---|---|---|---|---|---|---|---|---|---|
| Color | Brightness | 0.83 | 5.54 | 2.80 | 0.14 | 1.53 | 0.18 | 1.38 | 8.23 | 3.87 | 0.07 | 1.48 | 0.21 |
| | Contrast | 0.99 | 7.86 | 3.60 | 0.17 | 1.71 | 0.17 | 1.42 | 10.71 | 5.07 | 0.07 | 1.65 | 0.22 |
| | Saturate | 0.67 | 4.94 | 2.43 | 0.12 | 1.22 | 0.14 | 0.93 | 6.72 | 3.31 | 0.06 | 1.33 | 0.18 |
| Blur | Defocus | 0.84 | 5.26 | 2.71 | 0.15 | 1.37 | 0.15 | 0.66 | 5.27 | 2.44 | 0.04 | 0.88 | 0.10 |
| | Gaussian | 0.94 | 5.81 | 2.92 | 0.16 | 1.56 | 0.18 | 0.78 | 5.85 | 2.73 | 0.05 | 1.04 | 0.14 |
| | Glass | 0.80 | 5.17 | 2.65 | 0.16 | 1.32 | 0.14 | 0.65 | 5.29 | 2.39 | 0.04 | 0.82 | 0.09 |
| | Motion | 1.51 | 15.10 | 6.81 | 0.18 | 2.50 | 0.35 | 1.62 | 14.66 | 6.85 | 0.08 | 1.60 | 0.28 |
| | Zoom | 2.28 | 27.88 | 9.52 | 0.28 | 3.74 | 0.41 | 2.68 | 34.06 | 11.99 | 0.14 | 2.84 | 0.50 |
| Noise | Gaussian | 6.49 | 29.22 | 14.81 | 0.41 | 6.56 | 0.80 | 5.25 | 43.33 | 25.43 | 0.20 | 3.64 | 0.71 |
| | Impulse | 5.98 | 37.32 | 19.16 | 0.43 | 8.11 | 0.88 | 6.73 | 59.86 | 33.16 | 0.22 | 4.43 | 0.75 |
| | Speckle | 3.73 | 29.39 | 12.22 | 0.35 | 5.68 | 0.57 | 4.86 | 51.12 | 26.11 | 0.18 | 3.17 | 0.64 |
| | Shot | 4.87 | 26.32 | 12.34 | 0.36 | 5.60 | 0.69 | 4.65 | 42.07 | 22.91 | 0.18 | 3.26 | 0.67 |
| Quality | Pixelate | 0.86 | 5.95 | 2.51 | 0.19 | 1.51 | 0.13 | 0.82 | 7.66 | 2.83 | 0.05 | 1.02 | 0.10 |
| | JPEG | 1.98 | 27.21 | 6.82 | 0.32 | 3.62 | 0.36 | 2.73 | 33.93 | 10.55 | 0.13 | 2.59 | 0.41 |
| | Elastic | 1.15 | 14.93 | 3.92 | 0.22 | 2.28 | 0.22 | 1.70 | 21.82 | 5.99 | 0.08 | 1.61 | 0.20 |
| Weather | Fog | 2.35 | 15.39 | 10.13 | 0.19 | 2.43 | 0.19 | 2.29 | 18.15 | 11.67 | 0.06 | 1.23 | 0.15 |
| | Frost | 7.91 | 41.60 | 23.41 | 0.38 | 6.55 | 0.78 | 7.49 | 45.07 | 24.26 | 0.16 | 3.75 | 0.52 |
| | Rain | 10.21 | 41.78 | 28.99 | 0.70 | 12.79 | 1.29 | 27.89 | 74.23 | 59.77 | 0.47 | 10.75 | 1.96 |
| | Snow | 6.36 | 47.06 | 33.55 | 0.46 | 7.67 | 0.80 | 19.08 | 80.49 | 60.01 | 0.31 | 6.79 | 0.84 |
| | Spatter | 7.00 | 46.35 | 22.10 | 0.39 | 6.21 | 0.80 | 7.06 | 55.55 | 25.80 | 0.17 | 3.82 | 0.53 |
| Average | | 3.39 | 22.00 | 11.17 | 0.29 | 4.20 | 0.46 | 5.03 | 31.20 | 17.36 | 0.14 | 2.89 | 0.46 |
| Std. Dev. | | 2.95 | 15.23 | 9.60 | 0.15 | 3.11 | 0.34 | 6.85 | 24.26 | 17.63 | 0.11 | 2.40 | 0.43 |
| Median | | 2.13 | 20.86 | 8.17 | 0.25 | 3.06 | 0.35 | 2.49 | 27.88 | 11.11 | 0.10 | 2.12 | 0.35 |
| Clean Error | | 2.52 | 13.96 | 6.89 | 7.11 | 32.95 | 14.54 | 2.53 | 20.98 | 8.48 | 8.08 | 57.03 | 21.54 |

(b) Initial stereo disparity evaluation.

| | | RAFT-Stereo (s) | | | ACVNet (s) | | | LEAStereo (s) | | | LEAStereo (k) | | | GANet (k) | | | GANet (s) | | |
| | | $R^c_{\text{1px}}$ | $R^c_{\text{Abs}}$ | $R^c_{\text{D1}}$ | $R^c_{\text{1px}}$ | $R^c_{\text{Abs}}$ | $R^c_{\text{D1}}$ | $R^c_{\text{1px}}$ | $R^c_{\text{Abs}}$ | $R^c_{\text{D1}}$ | $R^c_{\text{1px}}$ | $R^c_{\text{Abs}}$ | $R^c_{\text{D1}}$ | $R^c_{\text{1px}}$ | $R^c_{\text{Abs}}$ | $R^c_{\text{D1}}$ | $R^c_{\text{1px}}$ | $R^c_{\text{Abs}}$ | $R^c_{\text{D1}}$ |
|---|---|---|---|---|---|---|---|---|---|---|---|---|---|---|---|---|---|---|---|
| Color | Brightness | 8.98 | 2.13 | 2.83 | 19.82 | 6.89 | 8.80 | 6.38 | 1.27 | 1.78 | 11.57 | 2.02 | 3.73 | 12.46 | 2.48 | 4.61 | 10.74 | 2.11 | 3.39 |
| | Contrast | 14.04 | 2.62 | 3.81 | 19.33 | 8.34 | 9.88 | 19.00 | 3.33 | 6.45 | 18.23 | 2.86 | 5.63 | 18.02 | 2.72 | 5.49 | 23.14 | 3.94 | 6.74 |
| | Saturate | 7.54 | 0.74 | 0.95 | 8.12 | 3.18 | 3.79 | 6.43 | 1.24 | 1.71 | 13.57 | 3.05 | 4.64 | 16.69 | 3.53 | 5.77 | 13.53 | 2.70 | 3.86 |
| Blur | Defocus | 10.61 | 2.47 | 3.90 | 8.06 | 1.10 | 1.90 | 8.55 | 2.02 | 2.49 | 29.31 | 3.26 | 5.21 | 41.32 | 3.29 | 4.68 | 12.34 | 2.46 | 3.16 |
| | Gaussian | 11.40 | 2.57 | 3.97 | 9.29 | 1.55 | 2.38 | 9.64 | 2.16 | 2.65 | 48.95 | 3.68 | 5.54 | 47.97 | 3.55 | 4.98 | 13.76 | 2.69 | 3.45 |
| | Glass | 13.10 | 2.61 | 3.34 | 11.72 | 1.31 | 1.95 | 11.56 | 2.17 | 2.55 | 70.01 | 4.79 | 6.36 | 71.45 | 4.33 | 5.18 | 19.42 | 2.61 | 3.15 |
| | Motion | 12.41 | 2.30 | 2.61 | 9.72 | 1.13 | 2.07 | 10.59 | 1.82 | 2.74 | 20.04 | 2.44 | 4.77 | 16.99 | 2.27 | 4.28 | 13.12 | 2.31 | 3.61 |
| | Zoom | 59.50 | 5.86 | 7.19 | 64.76 | 6.43 | 9.32 | 63.52 | 6.38 | 9.74 | 74.92 | 8.84 | 16.83 | 74.29 | 8.18 | 14.80 | 59.89 | 7.29 | 11.21 |
| Noise | Gaussian | 40.76 | 20.44 | 24.16 | 56.40 | 39.19 | 37.76 | 80.74 | 80.89 | 62.28 | 65.13 | 15.23 | 24.53 | 49.20 | 7.90 | 13.17 | 85.78 | 33.35 | 45.02 |
| | Impulse | 44.79 | 21.66 | 27.99 | 69.34 | 53.14 | 49.67 | 85.39 | 85.24 | 65.42 | 69.03 | 17.24 | 25.47 | 51.64 | 8.18 | 12.70 | 85.00 | 38.94 | 50.45 |
| | Speckle | 42.58 | 13.64 | 21.85 | 71.99 | 63.51 | 57.36 | 84.06 | 84.54 | 65.37 | 66.23 | 15.68 | 24.31 | 55.36 | 7.64 | 13.63 | 83.70 | 29.65 | 41.90 |
| | Shot | 39.84 | 15.55 | 20.23 | 59.56 | 42.20 | 41.10 | 79.41 | 76.53 | 59.94 | 64.06 | 14.29 | 22.95 | 49.36 | 6.95 | 11.98 | 81.49 | 28.20 | 39.89 |
| Quality | Pixelate | 66.69 | 46.19 | 13.86 | 57.29 | 4.14 | 4.98 | 35.19 | 3.85 | 4.11 | 57.19 | 3.72 | 4.83 | 62.71 | 4.00 | 4.60 | 59.61 | 3.70 | 4.07 |
| | JPEG | 55.27 | 8.24 | 5.27 | 60.87 | 15.98 | 15.16 | 55.18 | 9.20 | 10.84 | 68.22 | 5.63 | 7.97 | 65.92 | 7.41 | 11.19 | 59.52 | 6.76 | 10.10 |
| | Elastic | 65.53 | 6.52 | 4.32 | 58.39 | 8.17 | 7.29 | 71.96 | 8.02 | 10.92 | 93.40 | 7.16 | 8.90 | 87.38 | 6.89 | 8.86 | 76.47 | 4.85 | 5.05 |
| Weather | Fog | 13.71 | 1.57 | 2.10 | 17.99 | 17.70 | 12.12 | 17.95 | 14.25 | 10.88 | 23.36 | 8.18 | 12.90 | 21.36 | 9.69 | 12.45 | 20.55 | 9.68 | 9.75 |
| | Frost | 41.63 | 18.84 | 10.68 | 39.79 | 8.15 | 19.27 | 38.43 | 7.28 | 18.51 | 53.98 | 12.37 | 23.89 | 39.74 | 9.84 | 20.93 | 47.40 | 11.20 | 24.31 |
| | Rain | 43.10 | 79.42 | 32.27 | 34.62 | 12.92 | 18.48 | 56.55 | 22.14 | 34.58 | 65.45 | 12.54 | 28.62 | 49.08 | 11.44 | 22.55 | 59.22 | 26.50 | 42.34 |
| | Snow | 41.05 | 51.30 | 32.90 | 40.96 | 18.62 | 29.03 | 47.03 | 20.51 | 32.23 | 52.16 | 13.88 | 29.40 | 35.16 | 11.83 | 22.94 | 45.88 | 17.24 | 33.30 |
| | Spatter | 35.50 | 27.17 | 12.57 | 18.01 | 2.18 | 3.85 | 31.43 | 5.13 | 10.19 | 35.54 | 7.93 | 14.24 | 28.00 | 6.75 | 12.42 | 34.58 | 6.04 | 13.86 |
| Average | | 33.40 | 16.57 | 11.84 | 36.80 | 15.79 | 16.81 | 40.95 | 21.90 | 20.77 | 50.02 | 8.24 | 14.04 | 44.71 | 6.44 | 10.86 | 45.26 | 12.17 | 17.93 |
| Std. Dev. | | 20.16 | 20.72 | 10.79 | 23.56 | 18.64 | 17.08 | 29.19 | 31.33 | 23.64 | 23.69 | 5.15 | 9.43 | 21.37 | 3.00 | 6.10 | 28.09 | 12.17 | 17.25 |
| Median | | 40.30 | 7.38 | 6.23 | 37.21 | 8.16 | 9.60 | 36.81 | 6.83 | 10.51 | 55.58 | 7.55 | 10.90 | 48.53 | 6.92 | 11.58 | 46.64 | 6.40 | 9.93 |
| Clean | | 15.27 | 3.02 | 5.35 | 14.77 | 1.52 | 5.35 | 19.89 | 3.88 | 9.19 | 47.50 | 6.15 | 17.16 | 27.91 | 5.29 | 11.56 | 23.22 | 4.59 | 10.39 |

nificant performance losses for severe weather like rain and noise-based corruptions, *e.g.* impulse noise. Interestingly, their robustness does not improve compared to conventional optical flow models. Noise and weather corruptions remain a challenge for Disparity 2 predictions. Here, RAFT-3D consistently achieves lower robustness scores compared to M-FUSE, but conditions like impulse noise or rain still notably affect disparity predictions. Overall, both models have limited robustness, but temporal consistency may contribute to lower robustness scores under several corruption types.

**Stereo.** Results of stereo disparity estimations are presented in Tab. 2b. The effect of the different corruptions on the performance is significant, with noise and weather-based corruptions leading to the largest errors, especially for GANet and LEAStereo. In particular, Gaussian and impulse noise introduce extremely large errors, highlighting the sensitivity of stereo models to pixel-level noise. Blur distortions, especially zoom blur, also have a severe impact on all models. In contrast, color-based distortions generally yield smaller errors. RAFT-Stereo shows stronger resilience across most corruption groups, performing better on color and noise based corruption than other models, but also struggles with noise and severe weather effects such as rain and snow.

Table 3: Evaluations of the metrics used in RobustSpring.

(a) Influence of subsampling. We compare robustness evaluations on the full test data (Full) to evaluations on Spring's original subsampling (Spring), original subsampling without Hero-frames (Spring*), and our refined corruption subsampling (Ours).

(b) Robustness ranking of optical flow models with ranking strategies Average $R_{\text{EPE}}^c$, Median $R_{\text{EPE}}^c$, and Schulze to summarize results over corruptions. Please note that Schulze does not produce numeric values.

| | Subsampling $R_{\text{EPE}}^c$ | | | | Subsampling $R_{\text{1px}}^c$ | | | |
|---|---|---|---|---|---|---|---|---|
| | Full | Spring | Spring* | Ours | Full | Spring | Spring* | Ours |
| % Original Data | 100% | 1.00% | 0.94% | 0.05% | 100% | 1.00% | 0.94% | 0.05% |
| SEA-RAFT | 2.96 | 3.19 | 2.96 | 2.96 | 17.52 | 18.44 | 17.52 | 17.52 |
| GMFlow | 2.98 | 3.20 | 2.98 | 2.98 | 40.89 | 41.99 | 40.89 | 40.89 |
| MS-RAFT+ | 3.62 | 3.84 | 3.62 | 3.62 | 23.38 | 24.44 | 23.39 | 23.39 |
| FlowFormer | 3.77 | 3.89 | 3.77 | 3.77 | 21.52 | 22.39 | 21.53 | 21.53 |
| GMA | 4.03 | 4.28 | 4.03 | 4.03 | 21.47 | 22.59 | 21.48 | 21.47 |
| SPyNet | 4.30 | 4.56 | 4.29 | 4.29 | 38.32 | 39.28 | 38.32 | 38.32 |
| RAFT | 5.64 | 6.15 | 5.64 | 5.64 | 20.17 | 21.20 | 20.18 | 20.18 |
| FlowNet2 | 7.01 | 7.36 | 7.01 | 7.01 | 18.84 | 19.79 | 18.84 | 18.84 |
| PWCNet | 7.25 | 7.52 | 7.25 | 7.25 | 31.71 | 32.55 | 31.72 | 31.71 |

| | Ranking Method | | | | |
|---|---|---|---|---|---|
| Rank | Average $R_{\text{EPE}}^c$ | | Median $R_{\text{EPE}}^c$ | | Schulze |
| 1 | 2.96 | SEA-RAFT | 1.20 | SEA-RAFT | SEA-RAFT |
| 1 | 2.98 | GMFlow | 1.39 | GMA | MS-RAFT+ |
| 2 | 3.62 | MS-RAFT+ | 1.47 | FlowNet2 | GMA |
| 3 | 3.77 | FlowFormer | 1.71 | MS-RAFT+ | FlowNet2 |
| 4 | 4.03 | GMA | 1.92 | GMFlow | GMFlow |
| 5 | 4.29 | SPyNet | 2.14 | FlowFormer | FlowFormer |
| 6 | 5.64 | RAFT | 2.60 | RAFT | SPyNet |
| 7 | 7.01 | FlowNet2 | 2.77 | PWCNet | PWCNet |
| 8 | 7.25 | PWCNet | 2.82 | SPyNet | RAFT |

## 4.2 METRICS AND BENCHMARK CAPABILITY

After reporting initial RobustSpring results, we analyze aspects of its benchmark character: The subsampling strategy for data efficiency, and different ranking systems for result comparisons across 20 different prompt variations. We also validate our robustness metric for object corruptions and explore RobustSpring's transferability to the real-world.

**Subsampling.** We evaluate RobustSpring's strict data subsampling by comparing to results on the full test set. As shown in Tab. 3a, our subsampling strategy produces results that are nearly identical to those that include all pixels in the robustness calculation. We observe the largest discrepancy for Spring's original subsampling, because it includes a handful of full-resolution Hero-frames. If those frames are also subsampled (Spring*), results align with the full dataset. Overall, our stricter subsampling to 0.05% of all data is not only data efficient but also exact.

**Metric Ranking.** To explore how ranking strategies influence the optical-flow robustness order, we contrast our three summarization strategies: Average, Median, and Schulze (*cf.* App. A.4.3). The rankings in Tab. 3b notably differ across strategies. The Average differs most from the other rankings. For example, it ranks GMFlow 2nd, which is only 5th on Median and Schulze, suggesting a good performance across corruptions without excessive outliers but no top performance on most corruptions. Interestingly, Median and Schulze rankings are more aligned. As Schulze's ranking involves complex comparisons of per-corruption rankings and must be globally recomputed for new models, the Median ranking is a cheap approximation to it. The ranking strategy has significant implications for selecting robust models. While SEA-RAFT is optimal across rankings, the rankings accentuate different aspects: overall performance, outlier robustness, or balanced performance in pairwise comparisons. Hence, RobustSpring reports them all.

**Robustness on Object Corruptions.** In RobustSpring, robustness is defined as *stability* of predictions under corrupted inputs: models are robust if their outputs remain consistent for clean and

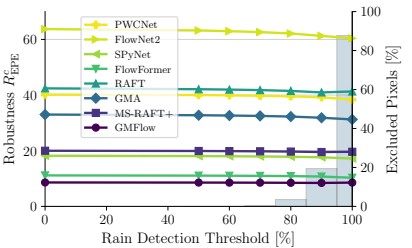 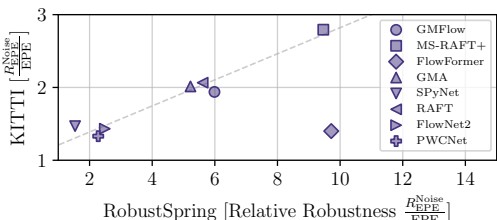

(a) Stability of corruption robustness $R_{\text{EPE}}^c$ on rain corruption. Robustness scores and rankings remain stable even if no rain pixels are in the $R_{\text{EPE}}^c$ calculation.

(b) Relative robustness to noise on RobustSpring transfers to noisy real-world KITTI data (Menze & Geiger, 2015) for most optical flow models.

Figure 5: Additional evaluations of RobustSpring's benchmark character.

corrupted scenes (Eq. 1). A natural question is whether this metric remains valid when corruptions introduce moving objects, such as rain or snow. To test this, we separate the contributions of background and corruption pixels by excluding pixels of objects like rain drops from the score calculation. Object pixels are detected via the value difference $d$ between clean and corrupted images, and excluded if $(1 - d)$ exceeds a threshold. Threshold 0 excludes none (the vanilla $R_{\text{EPE}}^{\text{Rain}}$), while threshold 100 excludes all. Figure 5a shows the robustness score if rain is excluded from the calculation, along with bars indicating the amount [%] of excluded pixels. Remarkably, the robustness score changes by at most $5\%$ even when all rain pixels (about $90\%$ of the image) are discarded. High scores for rain or snow (*cf.* App. A.4.4) thus result mainly from mispredictions in the *periphery* of altered pixels, not from motion predictions on altered pixels. As scene-wide effects dominate, our stability-based robustness yields consistent rankings suited for broad robustness evaluations.

**Robustness in the Real World.** Finally, we investigate if RobustSpring's corruption robustness transfers to the real world. To this end, we select the noisiest 10% KITTI data, estimating noise as in Immerkaer (1996). These noisy KITTI frames have no clean counterparts to calculate corruption robustness $R_{\text{EPE}}^{\text{Noise}}$. Thus, we approximate $R_{\text{EPE}}^{\text{Noise}}$ via the accuracy difference on noisy and non-noisy KITTI frames. To account for model-specific performance differences on Spring and KITTI, we normalize with the clean dataset performance and show the resulting relative robustness $\frac{R_{\text{EPE}}^{\text{Noise}}}{\text{EPE}^{\text{Clean}}}$ in Fig. 5b. Relatively robust models with low scores on RobustSpring are also robust on KITTI and vice versa. The only outlier, FlowFormer, overperforms on KITTI, potentially due to outstanding memorization capacity and exposure to KITTI during training. Because overall noise resilience on RobustSpring qualitatively transfers to KITTI, RobustSpring supports model selection for real-world settings where corruption robustness cannot be measured.

## 5 CONCLUSION

With RobustSpring we introduce an image corruption dataset and benchmark that evaluates the robustness of optical flow, scene flow, and stereo models. We carefully design 20 different image corruptions and integrate them in time, stereo, and depth for a holistic evaluation of dense matching tasks. Furthermore, we establish a corruption robustness metric using clean and corrupted predictions, and compare ranking strategies to unify model results across all 20 corruptions. RobustSpring's benchmark further supports data-efficient result uploads to an evaluation server. Our initial evaluation of 17 optical flow, scene flow, and stereo models reveals an overall high sensitivity to corrupted images. As our robustness results translate to real-world performance, systematic corruption benchmarks like RobustSpring are crucial to uncover potential model performance improvements.

**Limitations.** Due to its benchmark character, we have limited the image corruptions on RobustSpring to a selection of 20. While this does not cover the full space of potential corruptions, this data-budget limitation is necessary to make the RobustSpring dataset applicable and not overburden the computational resources of researchers during evaluation.

ACKNOWLEDGEMENTS

Lukas Mehl, Jenny Schmalfuss, and Victor Oei acknowledge funding by the Deutsche Forschungs-gemeinschaft (DFG, German Research Foundation) – Project-ID 251654672 – TRR 161: Quantitative Methods for Visual Computing (B04, A07). Jenny Schmalfuss and Andres Bruhn acknowledge support by the Deutsche Forschungsgemeinschaft (DFG, German Research Foundation) – Project-ID 533085500 - Robust Optical Flow. Shashank Agnihotri and Margret Keuper acknowledge support by the DFG Research Unit 5336 – Learning to Sense (L2S). Jenny Schmalfuss and Victor Oei acknowledge support from the International Max Planck Research School for Intelligent Systems (IMPRS-IS).

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

## A    APPENDIX

### A.1    DISCUSSION OF ROBUSTNESS DEFINITION

**Scope of RobustSpring's robustness definition.** Robustness is a multifaceted concept. In Robust-Spring we focus on one specific and mathematically established perspective: robustness as prediction stability under input corruptions. This choice is motivated by its independence from ground-truth definitions of optical flow, and by its practical utility in assessing whether models maintain consistent scene estimates in the presence of visual disturbances. Other notions of robustness, such as those directly tied to ground truth, are equally valid but outside the scope of this benchmark.

**Accuracy vs. Robustness.** RobustSpring explicitly disentangles *accuracy* and *robustness*. Accuracy measures the deviation between model predictions and ground truth, which depends on the adopted definition of optical flow—here, the true 3D motion of visible surfaces projected into the 2D image plane (Horn & Schunck, 1981; Baker et al., 2011). Robustness, by contrast, is defined via Lipschitz continuity (Eq. 1) as the stability of predictions under corrupted inputs, independent of ground truth.

**Why Ground-truth-free Robustness.** Some works define robustness as the difference between predictions on corrupted images and the corresponding ground truth (Ranjan et al., 2019; Agnihotri et al., 2024c). This approach entangles robustness with accuracy and requires ambiguous judgments about how each corruption changes the motion field (*e.g.* elastic transforms, rain, snow). By adopting a ground-truth-free definition, RobustSpring avoids this ambiguity and aligns with established robustness literature (Schmalfuss et al., 2022b; Tsipras et al., 2019; Taori et al., 2020; Hein & Andriushchenko, 2017; Pauli et al., 2022).

**Perfect Accuracy vs. Perfect Robustness.** Under our definitions, a perfectly accurate method may achieve poor robustness scores for corruptions that genuinely alter the motion field (*e.g.* snow, elastic transform), because its predictions change in accordance with the altered scene. Conversely, a perfectly robust method (constant predictions) is maximally stable but inaccurate. This disentanglement is intentional: it reveals cases where accuracy and robustness align versus diverge. Such divergence highlights model behavior that plain accuracy metrics cannot capture.

**Practical Relevance.** As demonstrated in Fig. 5a, current models tend to propagate spurious foreground motions introduced by rain into the background, leading to degraded scene estimates. RobustSpring's robustness metric captures this instability and thus provides valuable insight into real-world performance. The combination of accuracy and robustness offers practitioners a two-axis evaluation, allowing trade-offs between precision and stability depending on the target application.

### A.2    LINKS AND CHECKPOINTS FOR EVALUATED MODELS

We evaluated the original, author provided optical flow, scene flow and stereo methods on the RobustSpring dataset. Tab. 4 reports the repositories and checkpoints for the optical flow, scene flow

Table 4: Repositories and checkpoints used for evaluating methods in RobustSpring. *The Mixed checkpoint MS-RAFT+ is pretrained on Chairs and Things and then fine-tuned on a mix of Sintel, Viper, KITTI 2015, Things, and HD1k.

| Method | Repository | Checkpoint |
|---|---|---|
| **Optical Flow** | | |
| RAFT | `https://github.com/princeton-vl/RAFT` | Sintel |
| PWCNet | `https://github.com/NVlabs/PWC-Net` | Sintel |
| GMFlow | `https://github.com/haofeixu/gmflow` | Sintel |
| GMA | `https://github.com/zacjiang/GMA` | Sintel |
| FlowNet2 | `https://github.com/NVIDIA/flownet2-pytorch` | Sintel |
| FlowFormer | `https://github.com/drinkingcoder/FlowFormer-Official` | Sintel |
| MS-RAFT+ | `https://github.com/cv-stuttgart/MS_RAFT_plus` | Mixed* |
| SEA-RAFT | `https://github.com/princeton-vl/SEA-RAFT` | Spring (M) |
| SPyNet | `https://github.com/anuragranj/flowattack` (PyTorch implementation) | Sintel |
| | `https://github.com/anuragranj/spynet` (original implementation) | |
| **Scene Flow** | | |
| M-FUSE | `https://github.com/cv-stuttgart/M-FUSE` | KITTI 2015 |
| **Stereo** | | |
| RAFT-Stereo | `https://github.com/princeton-vl/RAFT-Stereo/` | Scene Flow (s) |
| ACVNet | `https://github.com/gangweiX/ACVNet` | Scene Flow (s) |
| LEAStereo | `https://github.com/XuelianCheng/LEAStereo` | Scene Flow (s), KITTI (k) |
| GANet | `https://github.com/feihuzhang/GANet` | Scene Flow (s), KITTI (k) |

and stereo models, which were benchmarked on RobustSpring in Tab. 1, Tab. 2a, and Tab. 2b. Further details on training and checkpoints for these models can be found in their original publications.

**Ressources and Run Times.** We conducted all evaluations on a single NVIDIA RTX A6000 (48 GB) GPU. The evaluation time of models on the RobustSpring data strongly depends on the computational efficiency and requirements of the original models. As a representative example, the optical flow evaluation with RAFT (Teed & Deng, 2020) took 20 hours on the full RobustSpring data.

Once the methods are evaluated and the results uploaded to our benchmark server prototype, the robustness evaluation for an optical flow method takes 13 minutes on our evaluation machine (Intel(R) Xeon(R) Gold 6130 CPU @ 2.10GHz, 8 vCPUs). The robustness evaluation for scene flow takes about 26 minutes, and 7 minutes for stereo.

## A.3    ROBUSTSPRING IMAGE CORRUPTIONS

Below, we provide supplementary information on the image corruptions for the RobustSpring dataset. Besides visualizing further benchmark samples and supplying a video that showcases the space- and time-integration of our corruptions, we also give details on their implementation with a focus on RobustSpring-specific consistencies.

### A.3.1    ADDITIONAL CORRUPTION BENCHMARK SAMPLES

To complement the benchmark samples in Fig. 3, we show benchmark results on additional corruptions in Fig. 6.

### A.3.2    VIDEO OF CORRUPTIONS

With the supplementary material, we include a video that visualizes all corruptions applied to the Spring test dataset.

### A.3.3    CORRUPTION IMPLEMENTATION

Below we provide the implementation details and parameters for all corruptions in the RobustSpring dataset. We cluster the corruptions by their main classes. The original (uncorrupted) image is denoted as $I$, while the corrupted version is $\hat{I}$. The pixel-aligned depth values are $D$. In the stereo video setting, the image subscripts $t$ and $t + 1$ denote frames over time, while $l$ and $r$ denote left and right frame, where necessary. All of RobustSpring's noises are independent of time, stereo and depth, which means they are sampled independently for every single image of the dataset.

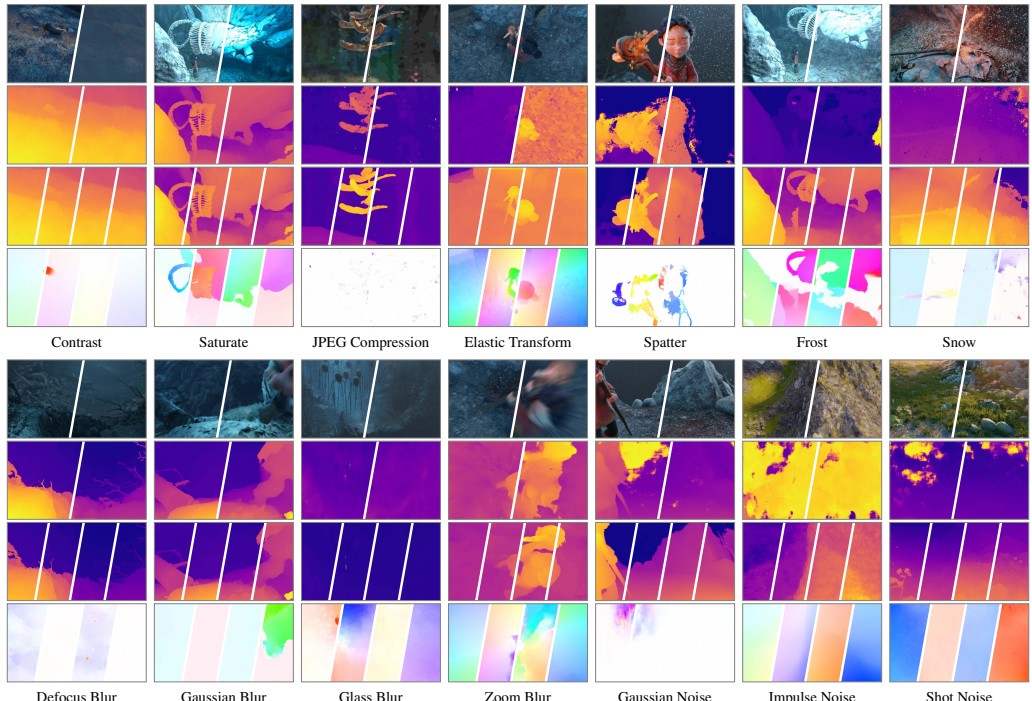

Figure 6: RobustSpring example frames, complementing Fig. 3. The first row shows clean and corrupted images. The second row shows the left and right disparity maps predicted with LEA Stereo (Cheng et al., 2020). The third row shows the target disparities for forward left, backward left, forward right, and backward right directions from M-FUSE (Mehl et al., 2023a). The fourth row shows optical flow estimates for forward left, backward left, forward right, and backward right from RAFT (Teed & Deng, 2020). All disparities and flows are computed on the corrupted dataset.

**Brightness.** The brightness is adapted via

$$\hat{I} = I + c, \tag{4}$$

and for time- and stereo-consistent brightness changes in RobustSpring we choose the parameter $c = c_t^l = c_t^r = c_{t+1}^l = c_{t+1}^r = 0.39$.

**Contrast.** The equation to adapt contrast is

$$\hat{I} = (I - mean(I)) \cdot c + mean(I), \tag{5}$$

where we selected $c = c_t^l = c_t^r = c_{t+1}^l = c_{t+1}^r = 0.16$ for time- and stereo-consistent contrast adaptations.

**Saturation.** For those adaptations the RGB image is transformed to HSV, and the saturation component $S$ is adapted via

$$\hat{S} = S \cdot \alpha + \beta, \tag{6}$$

with $\alpha = \alpha_t^l = \alpha_t^r = \alpha_{t+1}^l = \alpha_{t+1}^r = 2.3$ and $\beta_t^l = \beta_t^r = \beta_{t+1}^l = \beta_{t+1}^r = 0.01$ for time- and stereo-consistent saturation changes.

**Defocus Blur.** The defocus blur convolves the image with a circular mean filter $C^{mean}$

$$\hat{I} = I * C_r^{mean}, \tag{7}$$

where we choose the radius $r = r_t^l = r_t^r = r_{t+1}^l = r_{t+1}^r = 6$ for time- and stereo-consistent blurring.

**Gaussian Blur.** Gaussian blur convolves the image with a Gaussian $C^{gauss}$

$$\hat{I} = I * C_\sigma^{Gauss}, \tag{8}$$

where we choose the standard deviation $\sigma = \sigma_t^l = \sigma_t^r = \sigma_{t+1}^l = \sigma_{t+1}^r = 4$ for time- and stereo-consistent blurring.

**Glass Blur.** This is a Gauss-blurred image, whose pixels are afterwards shuffled via the shuffling $S(I, i, r)$ over several iterations $i$ within a neighborhood of radius $r$

$$\hat{I} = S(I * C_\sigma^{Gauss}, i, r), \tag{9}$$

where two sets of time-consistent parameters are picked for the different stereo cameras: $\sigma_t^l = \sigma_{t+1}^l = 1.2$, $\sigma_t^r = \sigma_{t+1}^r = 1.2$, $i_t^l = i_{t+1}^l = 1$, $i_t^r = i_{t+1}^r = 1$, $r_t^l = r_{t+1}^l = 3$ and $r_t^r = r_{t+1}^r = 3$.

**Motion Blur.** Motion blur is implemented by averaging the intensities of pixels along the motion trajectory determined by the optical flow. Let $\mathbf{v}(x, y) = (v_x(x, y), v_y(x, y))$ be the optical flow vector at pixel $(x, y)$, and let

$$N = \max\left(1, \left\lfloor 10 \cdot \max_{(x,y)} \|\mathbf{v}(x, y)\| \right\rfloor\right) \tag{10}$$

be the number of samples along the motion path. Then, the blurred pixel is computed as

$$\hat{I}(x, y) = \frac{1}{N+1} \sum_{k=0}^{N} I\left(x + \frac{k}{N} v_x(x, y), \, y + \frac{k}{N} v_y(x, y)\right). \tag{11}$$

Here, the scaling factor 10 controls the extent of the blur relative to the magnitude of the motion.

**Zoom Blur.** Zoom blur is created by averaging the original image with a series of zoomed-in versions of itself. Specifically, let $Z(I, z)$ denote the image $I$ zoomed by a factor $z$, and let $\{z_i\}$ be a set of zoom factors ranging from 1 to approximately 1.24 (in increments of 0.02). Then the final image is computed as

$$\hat{I} = \frac{1}{N+1}\left(I + \sum_{i=1}^{N} Z(I, z_i)\right), \tag{12}$$

where $N$ is the number of zoom factors. This formulation averages the original image with its progressively zoomed versions, resulting in a smooth zoom blur effect.

**Gaussian Noise.** This noise adds a random value from a Normal distribution to every pixel in the original image, where $\mathcal{N}_I(\mu, \sigma^2)$ is a $I$-shaped array of random numbers that are drawn from the Normal distribution with mean $\mu$ and variance $\sigma^2$:

$$\hat{I} = I + \alpha \cdot \mathcal{N}_I(0, 1). \tag{13}$$

The scaling $\alpha = 0.115$ is selected for all images in RobustSpring, but $\mathcal{N}_I(0, 1)$ is sampled anew for every image.

**Impulse Noise.** Here, for a fixed fraction of pixels $p$, their values are replaced by the values 0 or 255. For RobustSpring, $p = 0.075$.

**Speckle Noise.** Like Gaussian noise, Speckle noise also builds on random values from a Normal distribution, but adds these values after additionally scaling with $I$:

$$\hat{I} = I + I \cdot \alpha \cdot \mathcal{N}_I(0, 1). \tag{14}$$

For RobustSpring, the parameter is $\alpha = 0.45$.

**Shot Noise.** Shot noise uses values drawn from a Poisson distribution $\mathcal{P}$ per pixel

$$\hat{I} = \frac{\mathcal{P}(I \cdot c)}{c}, \tag{15}$$

where $c = 23$ for RobustSpring.

**Pixelation.** This is achieved by downsampling the image to size $s$, a fraction of its original size, with a box filter $box(I, s)$, followed by upsampling $up(I, s)$ to the size $s$, which is the original size:

$$\hat{I} = up(box(I, I \cdot c), I). \tag{16}$$

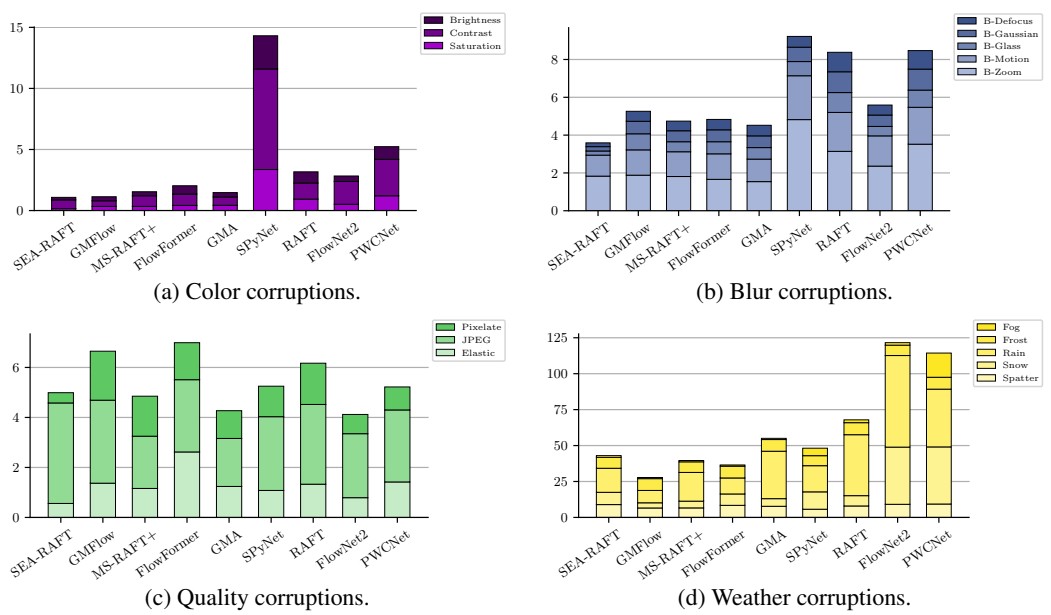

Figure 7: Additional results on accumulated corruption robustness $R_{\text{EPE}}^c$ for optical flow models over corruption classes color, blur, quality, and weather. More results are in Fig. 4.

For RobustSpring, we use the size fraction $c = c_t^l = c_t^r = c_{t+1}^l = c_{t+1}^r = 0.16$ for time- and stereo-consistent pixelation.

**JPEG Compression.** For JPEG compression, the quality $q$ is the only variable parameter

$$\hat{I} = JPEG(I, q), \tag{17}$$

which is selected as $q = q_t^l = q_t^r = q_{t+1}^l = q_{t+1}^r = 6$ for time- and stereo-consistent JPEG compression.

**Elastic Transformation.** The elastic transformation applies an elastic deformation using `torchvision.transforms.v2` with parameters $\alpha = 110.0$ and $\sigma = 5.0$ to control the deformation magnitude and smoothness, while preserving the original frame dimensions.

**Spatter.** The spatter corruption simulates liquid droplets by generating a liquid layer from Gaussian noise, applying blur and thresholding, and blending it with the original image using a predefined color.

**Frost.** The frost corruption overlays a frost texture onto the image by randomly selecting and resizing a pre-stored frost image and blending it with the input to create an icy appearance.

**Snow and Rain.** The implementation for snow and rain is based on Schmalfuss et al. (2023), with methodological and performance improvements. On the methodological side, we replaced additive blending with order-independent alpha blending (Meshkin's Method, McGuire & Bavoil (2013)) and included global illumination (Halder et al., 2019) in the color rendering. Also, we expanded the monocular two-step motion simulation to multi-step stereo images. On the performance side, we introduce an efficient parallel particle initialization and improve the parallel processing performance.

**Fog.** Fog is modelled using the Koschmieder model from Wiesemann & Jiang (2016) as

$$\hat{I} = I \cdot e^{-\frac{D \cdot \ln(20)}{d_m}} + l \cdot \left(1 - e^{-\frac{D \cdot \ln(20)}{d_m}}\right), \tag{18}$$

where $d_m$ is the visibility range and $l$ the luminance of the sky. For RobustSpring, we use $d_m = 45$ and $l = 0.8$. As it directly depends on the depth $D$, this is depth-consistent and due to its integration into the 3D scene it is also stereo- and time-consistent.

## A.4 ADDITIONAL EXPERIMENTAL RESULTS

Below, we expand on the experiments in the main paper and provide supplementary results for our major experiments.

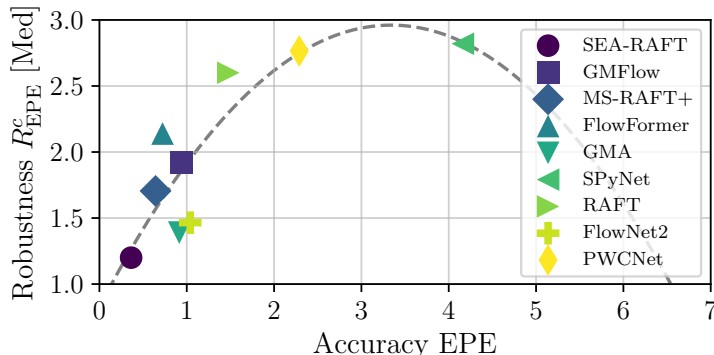

Figure 8: Accuracy vs. robustness of optical flow methods, measured as EPE and median $R_{\mathrm{EPE}}^c$. Small values indicate accurate and robust methods. The dashed line represents a quadratic polynomial fit. Fig. 4c shows the average $R_{\mathrm{EPE}}^c$.

Table 5: Pairwise comparison matrix for the Schulze method.

|  | SEA-RAFT | GMFlow | GMA | MS-RAFT+ | FlowFormer | SPyNet | RAFT | PWCNet | FlowNet2 |
|---|---|---|---|---|---|---|---|---|---|
| SEA-RAFT | 0 | 14 | 14 | 14 | 14 | 17 | 17 | 15 | 19 |
| GMFlow | 6 | 0 | 9 | 14 | 9 | 10 | 16 | 9 | 14 |
| MS-RAFT+ | 6 | 9 | 0 | 15 | 11 | 11 | 19 | 12 | 15 |
| FlowFormer | 6 | 6 | 5 | 0 | 3 | 12 | 16 | 8 | 13 |
| GMA | 6 | 11 | 9 | 17 | 0 | 12 | 20 | 10 | 15 |
| SPyNet | 3 | 10 | 9 | 8 | 8 | 0 | 13 | 4 | 13 |
| RAFT | 3 | 4 | 1 | 4 | 0 | 7 | 0 | 4 | 9 |
| FlowNet2 | 5 | 10 | 8 | 12 | 10 | 16 | 16 | 0 | 18 |
| PWCNet | 1 | 6 | 5 | 7 | 5 | 7 | 11 | 2 | 0 |

### A.4.1  CORRUPTION ROBUSTNESS BY CORRUPTION GROUP

Figure 7 shows the corruption robustness $R_{\mathrm{EPE}}^c$ for each optical flow method across the remaining four corruption groups in addition to Fig. 4 in the main paper. It underlines the varying degrees of robustness of the evaluated methods against specific types of corruption.

### A.4.2  ACCURACY VS. MEDIAN CORRUPTION ROBUSTNESS

In Fig. 8 we show the accuracy-robustness evaluation with the *Median* corruption robustness, to complement Fig. 4c which uses the average corruption robustness. Even though the robustness ranking of methods varies between average and media corruption robustness, *cf.* Sec. 4.2 for a discussion on ranking differences, the general trend that corruption robustness and accuracy are weakly correlated remains. However, there is still no clear winner, and an accuracy-robustness tradeoff persists among particularly accurate or robust methods.

### A.4.3  SCHULZE PAIRWISE COMPARISON MATRIX

The Schulze method is a ranking algorithm used to determine the most preferred candidate based on pairwise comparisons. We include a pairwise comparison matrix in Table 5 for our ranking. The table shows how often the method in row $i$ is better than the method in column $j$, based on the number of corruptions where method $i$ achieves a lower error than method $j$. The ranking process consists of the following steps:

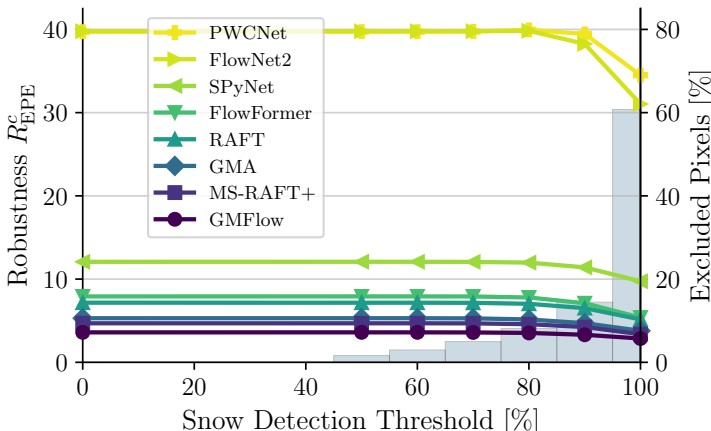

Figure 9: Stability of corruption robustness $R_{\mathrm{EPE}}^c$ for snow corruption. Analogous rain results in Fig. 5a.

**1) Constructing the Pairwise Comparison Matrix.** For each pair of methods, we count how many times one method achieves a lower EPE than the other across different corruptions. If method $A$ has a lower EPE than method $B$ in a given corruption, the corresponding entry in the matrix is incremented.

**2) Computing the Strongest Paths.** We define the strength of a path from method $A$ to method $B$ as the number of cases where $A$ outperforms $B$. The strongest paths between methods are determined by considering indirect paths: if method $A$ is better than method $B$, and method $B$ is better than method $C$, then the strength of the indirect path from $A$ to $C$ is considered.

**3) Determining the Final Ranking.** Method $A$ is ranked higher than method $B$ if the strongest path from $A$ to $B$ is stronger than the strongest path from $B$ to $A$. This ensures that even if a method loses to another in some comparisons, it can still be ranked higher if it consistently performs well against other methods.

### A.4.4 CORRUPTION ROBUSTNESS ON SNOW

Finally, we complement the evaluation of our corruption robustness metric in the presence of rain in Fig. 5a with the corresponding evaluation in the presence of snow in Fig. 9. The results with rain translate to snow with the following minor differences: Because snow has less motion blur than rain, it covers fewer pixels (60% of all pixels vs. 90% for rain). For snow, the score drops a bit more than for rain when object pixels are excluded ($\leq$25% drop vs. $\leq$5% for rain), potentially as a consequence of the increased object opacity for snow particles. Still, the background error ($\geq$ 75% contribution to corruption robustness) dominates the score, and the robustness ranking for optical flow methods remains stable, whether snow pixels are included in the score calculation or not. Hence, the additional evaluation on snow further substantiates the stability and expressiveness of corruption robustness as an evaluation metric.

### A.4.5 REPRODUCIBILITY

We compute the noise in the KITTI (Geiger et al., 2012) frames according to Immerkaer (1996), and select the top 10% noisiest frames. The final frames and their noise levels are listed in Tab. 6.

## A.5 DEPTH AND EXTRINSICS ESTIMATION FOR CORRUPTIONS

**Disparity and Depth.** For depth-dependent corruptions, we estimate disparity maps with MS-RAFT+ (Jahedi et al., 2022; 2024). On the Spring validation set, we find a mean EPE of 1.09 px and D1-all of 5.60%. Mean EPE measures the average Euclidean distance between predicted and ground-truth disparities, while D1-all reports the percentage of pixels with disparity error larger than 3 px or 5% of the ground-truth disparity. These errors are sufficiently small to support realistic

Table 6: Top 10% noisiest KITTI frames with noise estimation from Immerkaer (1996). Frames from kitti15/dataset/training/image_2.

| Frame | 000061_10.png | 000060_10.png | 000065_10.png | 000068_10.png | 000143_10.png | 000069_10.png | 000174_10.png | 000142_10.png | 000107_10.png | 000175_10.png | 000064_10.png | 000063_10.png | 000066_10.png | 000067_10.png | 000062_10.png | 000055_10.png | 000154_10.png | 000158_10.png | 000157_10.png | 000054_10.png |
|---|---|---|---|---|---|---|---|---|---|---|---|---|---|---|---|---|---|---|---|---|
| Noise | 3.54 | 3.22 | 3.07 | 2.92 | 2.80 | 2.79 | 2.77 | 2.74 | 2.69 | 2.66 | 2.62 | 2.61 | 2.61 | 2.60 | 2.48 | 2.47 | 2.46 | 2.46 | 2.43 | 2.41 |

augmentations. Note that depth estimates are used by only 3 of the 20 corruption types (fog, snow, and rain; motion blur uses depth only implicitly).

**Extrinsics.** We estimate camera extrinsics using COLMAP 3.8. As trajectories are not expressed in a common reference frame, quantitative pose metrics such as absolute trajectory error (ATE) or relative pose error (RPE) cannot be meaningfully computed without a rigid alignment step, which itself introduces error. Extrinsics are required for only 2 corruptions (snow, rain). We therefore evaluate them qualitatively: the resulting augmentations exhibit realistic motion behavior for both effects.

## A.6 User Study on SSIM Thresholds

The SSIM thresholds of 0.7 for color, blur, quality, and weather corruptions and 0.2 for noise were chosen empirically to yield comparable perceptual corruption strength across categories.

We conducted a focused perceptual study to validate the choice of corruption strengths and their associated SSIM values in RobustSpring. For five representative corruptions (one per main corruption family: contrast, fog, Gaussian blur, Gaussian noise, and pixelation), participants repeatedly viewed a clean Spring frame alongside three corrupted versions. See Figs. 10 and 11 for examples. They then selected the corrupted image that looked most like a "reasonably" corrupted version of the clean frame. This means the image is clearly degraded, inducing a distribution shift, yet still recognizably close to the original. Across 14 user studies, with 10 samples each, for a total of 140 comparisons, we observed agreement rates ranging from 70% to 100% from each user for the SSIM values used in our study. The average agreement was 87.14%, which suggests that the SSIM ranges used for these representative corruptions align with human perception of realistic, moderate corruption strengths.

In the user study, the web interface displayed one clean reference image from Spring and three corresponding corrupted versions of the same image on each page. These images were obtained by applying a single corruption type at three different intensities. We applied thresholds for three levels of severity: low, medium, and high, with corresponding SSIM values of 0.95, 0.7, and 0.2, respectively. The three corrupted images were randomly ordered, and participants were asked to select one image per page that they perceived as the most reasonably corrupted version of the clean image. The corruption should be clearly visible and sufficient to cause a distribution shift but not so strong that the scene appears implausible or too different from the original image. This setup was repeated for the five corruption types listed above, and for each page and participant, the selected intensity was recorded. This allowed us to compute per-participant agreement with the nominal level and the aggregate agreement statistics reported above. In each scenario, the nominal value corresponds to the SSIM value chosen for the respective corruption in our work.

## A.7 Motion Range of Spring

As discussed in the Spring paper (Mehl et al., 2023b), the dataset covers a wider range of motion than standard benchmarks. While Sintel (Butler et al., 2012) and KITTI (Menze & Geiger, 2015) align with Spring for smaller motions (u: $-500$ to $500$, v: $-250$ to $250$), Spring extends the range up to $1700$ (u) and $-750$ (v). This corresponds to motion magnitudes about 1.5–3 times larger than those in Sintel and KITTI. We highlight this here as one reason for selecting Spring as the foundation of RobustSpring.

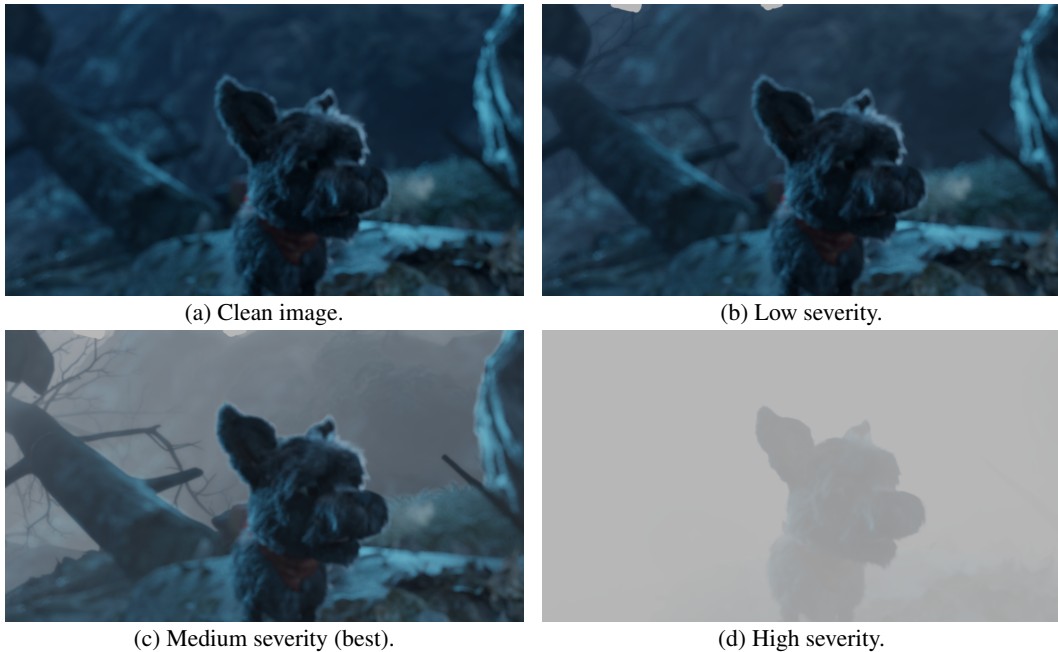

(a) Clean image.

(b) Low severity.

(c) Medium severity (best).

(d) High severity.

Figure 10: Clean image and three corrupted versions with different severity levels for fog used in the user study on SSIM thresholds. "Best" is the one we choose in RobustSpring.

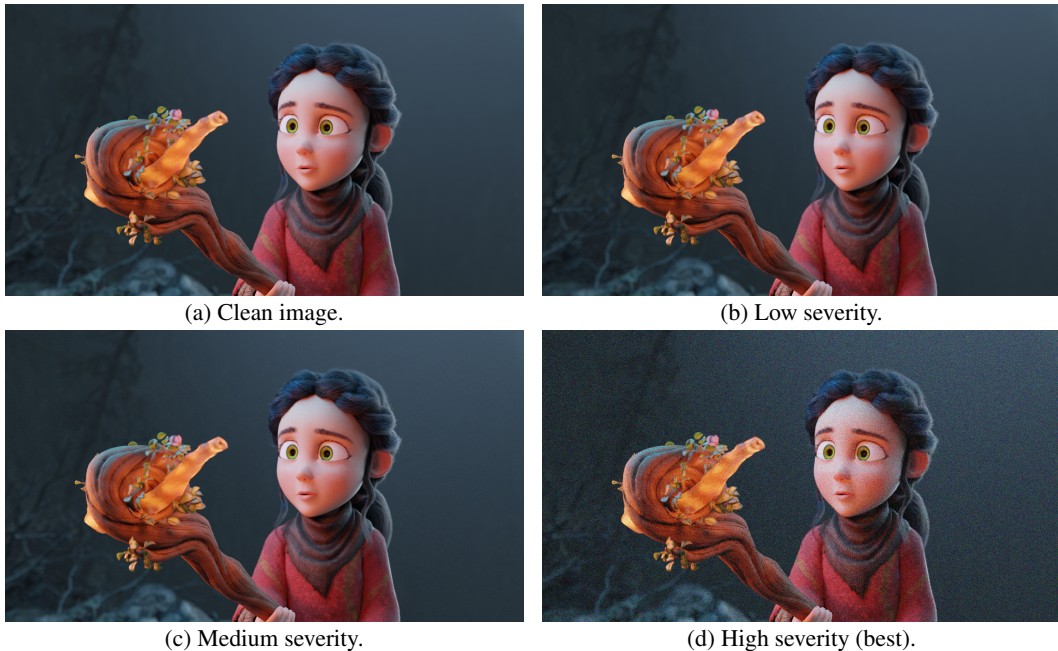

(a) Clean image.

(b) Low severity.

(c) Medium severity.

(d) High severity (best).

Figure 11: Clean image and three corrupted versions with different severity levels for Gaussian noise used in the user study on SSIM thresholds. "Best" is the one we choose in RobustSpring.

## A.8    INTEGRATION INTO THE SPRING BENCHMARK WEBSITE

To illustrate how RobustSpring extends the existing Spring benchmark infrastructure, we provide mock-ups of the website with our additional robustness results. These examples demonstrate how robustness values are integrated alongside existing accuracy metrics.

**Overview Page.** Fig. 12 shows the modified overview page for optical flow methods. Three new columns report the average robustness scores. This extension allows users to compare methods across both accuracy and robustness at a glance.

**Method Detail Page.** Fig. 13 displays the detail page for a single optical flow method. In addition to the accuracy results already present in Spring, robustness values are provided per corruption type, as well as aggregated average and median robustness values. This detailed view enables users to assess strengths and weaknesses of individual methods with respect to specific corruptions.

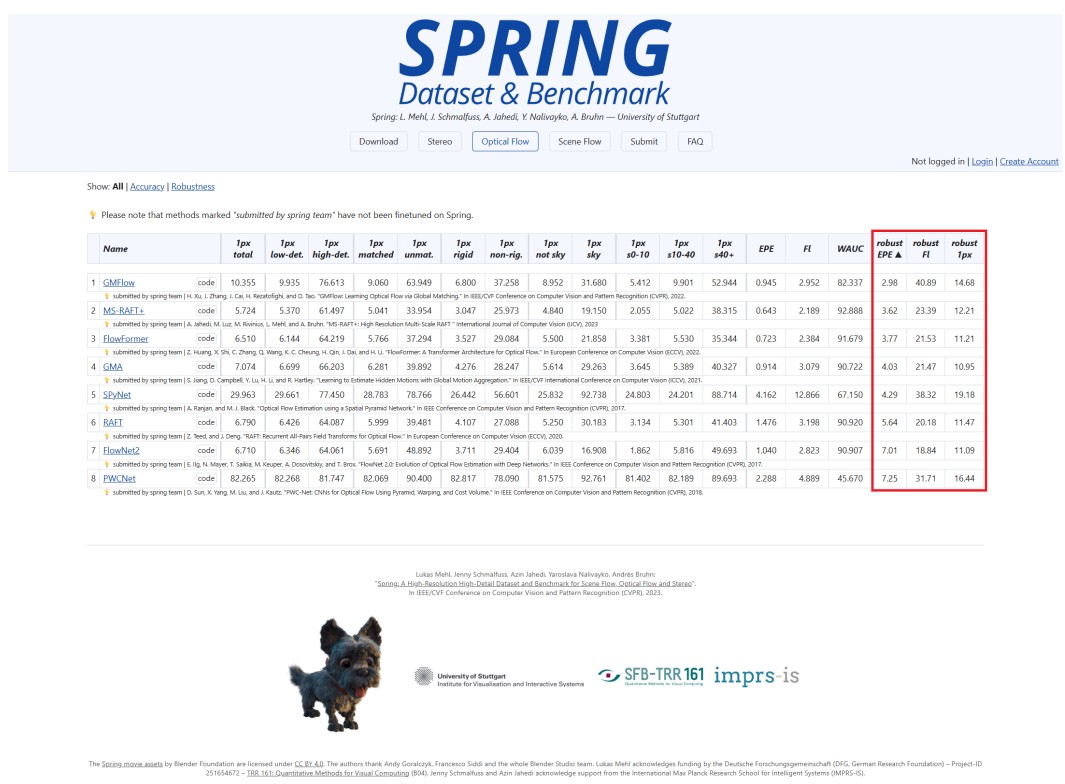

Figure 12: Modified Spring benchmark overview page for optical flow. Newly added robustness columns are highlighted in red.

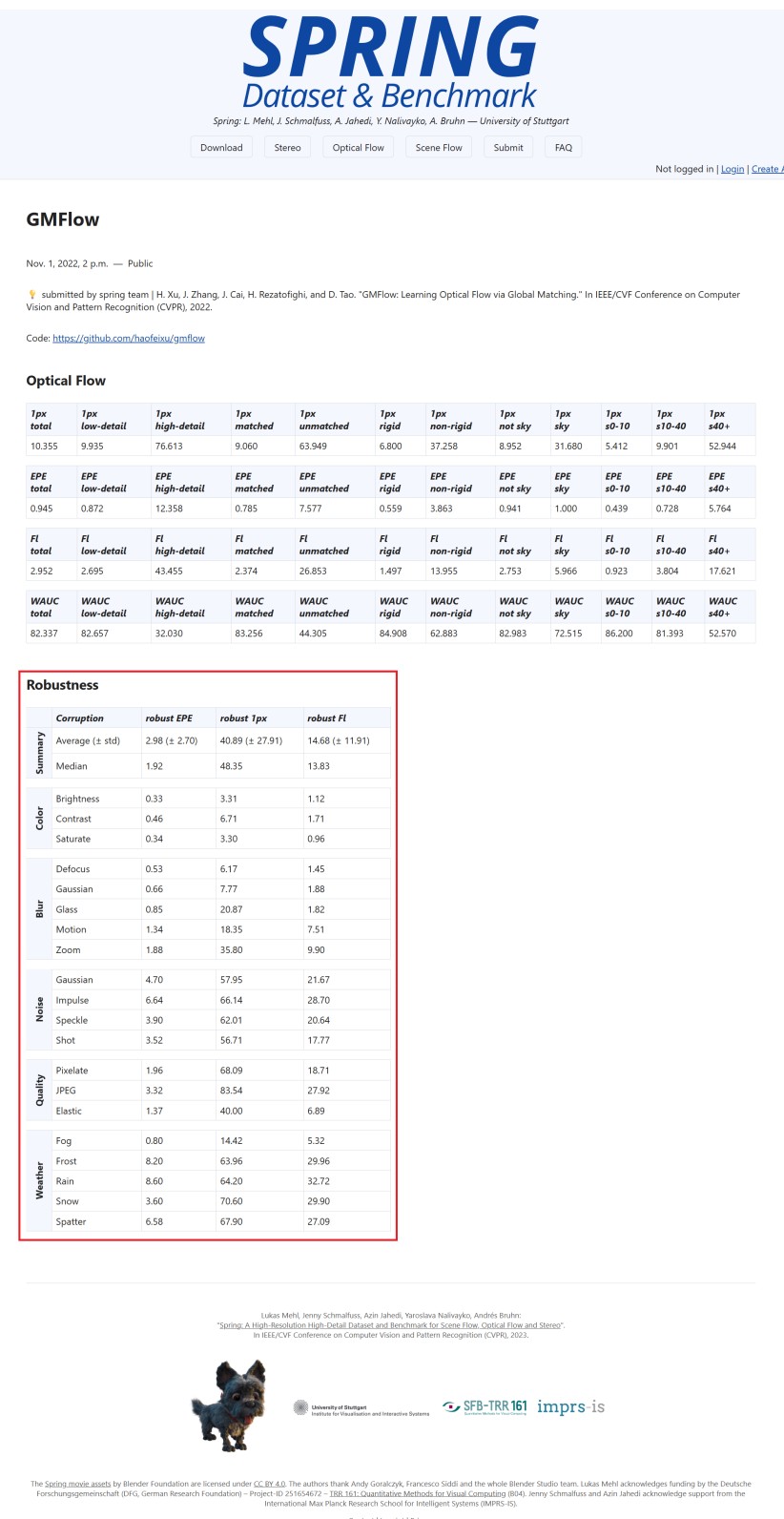

Figure 13: Modified Spring benchmark method detail page for optical flow. Robustness values are reported for each corruption, as well as aggregated average and median scores. Newly added sections are highlighted in red.

