# OpenReview forum: "RobustSpring: Benchmarking Robustness to Image Corruptions for Optical Flow, Scene Flow and Stereo"
_ICLR.cc/2026/Conference — ICLR 2026 Poster_

### Official Review · Reviewer_oJUR · 2025-10-26

**Soundness:** 3
**Presentation:** 3
**Contribution:** 3
**Rating:** 8
**Confidence:** 5

**Summary:**

The paper proposes a dataset and benchmarking methodology to assess the influence of image corruptions on optical flow estimation, disparity estimation, and scene flow estimation. Following upon similar assessments in the area of image classification, this type of analysis is extended here not only to dense prediction problems but also to correspondence estimation, which necessitates that (some types of) image corruptions are applied in a consistent manner across views. The analysis builds upon the recent Spring benchmark for optical flow estimation, and complements the accuracy evaluation of Spring with a robustness evaluation, which relies on estimating the stability of the prediction w.r.t. the perturbed image and does not rely on any ground truth. An evaluation of several standard models from the literature complements the analysis.

**Strengths:**

* A benchmark that assesses the robustness of motion and depth estimation methods w.r.t. image corruptions does not exist so far. This paper extends similar efforts in image classification to correspondence estimation methods and provides a novel and very useful framework to analyze the robustness of motion and depth estimation methods in detail.
* The benchmark is competently designed and well executed on top of one of the most prominent recent benchmark datasets for motion and depth estimation (Spring). I find it very sensible that robustness is decoupled from accuracy in the evaluation.
* The proposed benchmark has the potential to become a standard tool for analyzing the robustness of flow and depth estimation methods.
* Some useful insights emerge from the analysis, such as which family of methods perform well under which conditions. This can allow to design new architectures in the future.
* The paper is written quite well overall and is easy to follow. Occasionally, there are some odd phrases (see below), but these do not distract majorly from understanding the paper.

**Weaknesses:**

Major points:
* The paper does not argue very clearly why the specific set of image corruptions are particularly relevant in the context of motion and depth estimation. A clear justification why these are the prominent corruptions in real-world imagery would strengthen the paper.
* The notion of stereo, depth and time-consistent corruptions is not clearly defined in the main paper and only becomes more apparent from the supplemental material.
* I find the notion of stereo, depth, and time-consistent corruptions to be suggesting more than there actually is to it, and also possibly confusing. What the paper seems to want to say with this is that the same noise strength is applied to all views, but that’s not particularly surprising. What’s more: it is also possibly confusing because the reader may, for example, wonder what a stereo-consistent JPEG compression would be. Is it somehow a compression algorithm that considers two frames at once to adapt the compression? Yet, this is not actually the case: all this means is that the compression parameter is the same for both views, which again is sort of a given. The only place where the consistency is non-trivial seems to be the consistent synthesis of rain and snow, which is done in 3D. Details are largely absent though, also from the supplement.
* I find it surprising that SEA-RAFT (ECCV 2024) has not been included in the analysis. AFAIK, SEA-RAFT is the most competitive member of the RAFT family to date.
* I am somewhat surprised about the finding that the most accurate models are also the most robust (l. 370f). The paper argues in many places that accuracy and robustness do not necessarily go hand in hand, which I find plausible. Given this, I would have expected this surprising finding to be discussed more deeply.
* [This might be a contentious point, but one may wonder whether this paper is a good fit for ICLR. This is a pure computer vision paper without any learning component to it and thus would be a better fit for CVPR/ICCV/ECCV. That said, ICLR does include benchmarks in its call for papers and indeed the proposed benchmark allows to assess the robustness of learned motion and depth estimators. I am not taking this point into account in my rating and will leave the assessment of topical fit to the (S)AC.]

Minor points:
* The metrics in l. 300 should be explained, at least briefly.
* It does not become clear from the main paper why some frames (“Hero” frames) are being excluded. This becomes apparent only from an experiment in the supplement.
* Minor language oddities, e.g. l. 115 (models do not report, but papers do), l. 139 (evidence is not a verb), or l. 188 (what does it mean for all corruptions to be on a single frame?).

**Questions:**

* It would be good if the authors could discuss why they chose these particular robustness directions. Yes, they have mostly been considered for image classification, but is it obvious that these are the most important corruption dimensions to consider?
* Is the snow and fog synthesis really the only place where the consistency goes beyond using the same parameters for all views?
* Can the authors discuss the finding that accurate models are also robust in light of the desire to decouple the two? Would this analysis perhaps change if we not only normalized the difference in prediction w.r.t. the difference between the perturbed inputs but also w.r.t. the average displacement/disparity?
* I do not understand completely why the authors used estimated depth etc. to render 3D-consistent artifacts like rain. Put differently, why is leakage of ground truth a worry? In a real scene, rain would be consistent with the true 3D geometry of the scene, so why shouldn’t we use the ground truth depth for synthesizing rain? Sure, this would give some information about the true depth, but so would a capture of a scene with real rain.
* How are the target SSIM values of 0.7, respectively 0.2 chosen?
* What are “unoptimized data shifts”? (l. 113)
* What does the paper mean by “robustness to corruptions is undefined”? (l. 253) If this was really the case, how could this paper assess robustness to corruptions?
* Which norm is assumed in Eq. 1?
* Why does it make sense to omit the denominator from Eq. 2? The denominator in Eq. 1 is not based on the SSIM but rather on the norm of the image difference, and fixed a SSIM does not guarantee a certain norm on the image difference.

---

> ### Author Response · Authors · 2025-11-27
>
> Thank you for your thoughtful review and comments, and your very positive evaluation of our RobustSpring benchmark! We appreciate the recognition of RobustSpring as the first systematic benchmark for robustness in dense correspondence estimation and of its potential to become a standard tool in the community. Below we address each question in detail.
>
> **Choice and relevance of corruptions**
>
> We agree that this can be made more explicit. Please refer to comment (4) to reviewer gjCL for a more detailed discussion.
>
> **Stereo, depth, and time consistency**
>
> We appreciate the opportunity to clarify this terminology. By time consistency, we mean that the same corruption evolves smoothly across subsequent frames for one camera, as would occur with a persistent lens effect (e.g., frost, where the same frost pattern is applied over time to one camera, but the specific frost pattern differs among the stereo cameras). Stereo consistency means that both cameras experience the same photometric modification (e.g., identical brightness change), while depth consistency refers to corruptions rendered in 3D space (e.g., snow, rain, fog) that project coherently into both stereo views. Even though the noise parameters in the appendix are the same for all four considered frames, they are only descriptive of the noise intensity - however, the realization of the noise differs among all four frames. We will make this distinction clearer in the main paper and the supplementary. We agree that not all corruptions benefit from all three consistencies (only weather corruptions require explicit 3D rendering) and clarify this in the revised paper (Sec. 3.1, "Corruption Consistencies").
>
> **Relation between accuracy and robustness**
>
> Thank you for raising this point. Although Fig. 4c shows a mild trend that accurate models are also more robust to corruptions, this correlation is highly corruption-dependent. We added a more detailed discussion of accuracy–robustness relations to Sec. 4.1, Optical Flow.
>
> **Use of estimated depth for 3D corruptions**
>
> Leakage of ground truth depth is a potential issue because depth estimation is part of the original Spring benchmark, which withholds depths and extrinsics for the test set by design.
> Especially for corruptions like fog, where the color deviation of the corrupted image to the original image is a direct function of the depth, this would allow to reconstruct this withheld ground truth depth with very high accuracy.
> Using estimated geometry preserves benchmark integrity (for the original Spring accuracy benchmark) while still allowing realistic particle trajectories for rain and snow.
> Importantly, we have quantified the error that is introduced by estimating the depth fields, and the estimates are accurate enough for plausible 3D rendering (see App. A.5).
>
> **Target SSIM values and perceptual validation**
>
> The thresholds of 0.7 and 0.2 were empirically chosen to yield comparable perceptual corruption strength across categories.
> We conducted a user study to validate the choice of corruption strengths and their associated SSIM values in RobustSpring.
> Please refer to App. A.6 for more details.
>
> **Clarifications regarding Equations 1 and 2**
>
> The Lipschitz constant is generally defined using a valid metric for the respective metric space, and therefore can use different metrics for nominator and denominator in Eq. 1. Therefore, we refrained from chosing a specific metric in Eq. 1.
> Our use of Eq. 1 as a basis for Eq. 2 is conceptual: It motivates robustness as prediction stability under input perturbations. While the SSIM is not a norm, it has been shown that it can, under certain conditions, be converted into a distance measure [Brunet et al. "On the Mathematical Properties of the Structural Similarity Index", TPAMI 2012]. Here we make use of the fact that all models are evaluated on exactly the same corrupted image, which means the denominator $\|I-I_c\|$ is identical for all models within a corruption type and does therefore not influence the ranking among models. The use of the SSIM within this equalization is to ensure comparable perceptual severity, and not to provide a full approximation of the Lipschitz constant.
>
> **SEA-RAFT**
>
> Thank you for the suggestion. We added SEA-RAFT to the list of evaluated optical flow methods and updated the respective Tables and Figures.
>
> **Minor questions**
>
> - "Unoptimized data shifts" refers to domain shifts that occur naturally in data (e.g., noise, lighting) rather than those optimized adversarially against a specific model. We rephrased with "non-adversarial data shifts".
> - "Robustness to corruptions is undefined" refers to the lack of a standardized metric for dense matching. We rephrased this.
> - The metrics (EPE, 1px, Fl, Abs, D1) are now briefly explained in the main text.
> - The exclusion of hero frames is now clarified in Sec. 3.2. In Spring, the 10 hero frames are not subsampled to enable visualizations.
> - Minor language issues were corrected.

---

### Official Review · Reviewer_qH6X · 2025-10-27

**Soundness:** 2
**Presentation:** 3
**Contribution:** 2
**Rating:** 6
**Confidence:** 3

**Summary:**

This paper proposes RobustSpring, a new benchmark that extends Spring with various augmentations. The authors also propose novel corruption robustness metrics along with efficient subsampling methods for evaluation. Detailed experiments are provided.

**Strengths:**

RobustSpring introduces a large, diverse corruption benchmark for optical flow, scene flow, and stereo, enabling consistent robustness evaluation across real-world perturbations. It provides new metrics along with efficient subsampling evaluation, promoting robustness as a core research goal.

**Weaknesses:**

I describe my concerns in "Questions".

**Questions:**

I have several questions about the metrics and list them below.

- [Object Corruptions] What does "value difference" mean (Line 453)? Is that something similar to the intensity difference? Since the rain/snow effects are rendered in blender, why does the paper uses intensity difference instead of directly obtaining rain/snow pixels from rendered results?
- [Relative Robustness] What does this metric mean (Figure 5b)?
- How do corruptions affect the model performance? From Table 1 I can only tell whether a model is robust to different corruptions, but do these corruptions always make the predictions harder? If a corruption actually makes the prediction easier (e.g. better EPE), should we include this data point into proposed robustness metrics?

I'm happy to adjust my score if my concerns are addressed.

---

> ### Author Response · Authors · 2025-11-27
>
> Thank you for your constructive questions and the positive evaluation! We appreciate the recognition of RobustSpring's contributions in introducing a large-scale corruption benchmark and efficient evaluation protocol. Below, we address each of the concerns in detail.
>
> **(1) [Object Corruptions]**
>
> We apologize for the ambiguity. The value difference $d=I-I_c$ refers to the per-pixel intensity difference between the clean image $I$ and its corrupted version $I_c$, similar to the difference term used in the robustness metric (Eq. 1). We employ this intensity-based detection because the weather effects (i) were created in a post-processing step rather than directly in Blender and (ii) are composited with transparency and global illumination, such that the binary particle masks would not match the visible extent of the corruption in the final frame. Using the unified intensity difference provides a consistent measure of corruption-affected pixels across all cases. We clarified this in Sec. 3.2.
>
> **(2) [Relative Robustness]**
>
> The relative robustness metric in Fig. 5b is related to the real-world transfer experiment described in Sec. 4.2 ("Robustness in the Real World"), and is an approximation to RobustSpring's corruption robustness metric in the absence of clean and corrupted image versions. On KITTI, clean-corrupt image pairs do not exist, so the standard robustness measure $R_{\text{EPE}}$ cannot be computed. To approximate corruption sensitivity, we compare accuracy on the noisiest 10\% of KITTI frames to accuracy on the remaining frames, and normalize this degradation by the clean error.
> This metric therefore quantifies how much performance deteriorates under (real) noise, allowing us to test whether RobustSpring's robustness scores predict real-world behavior. We will add this additional explanation to the paper. As Fig. 5b shows, models that are robust on RobustSpring are also robust on KITTI, confirming the transferability of our benchmark.
>
> **(3) [How do corruptions affect the model performance?]**
>
> In principle, our corruption generation and SSIM equalization (Sec. 3.1) are designed such that all corruptions increase task difficulty. In practice, the corruption robustness metric $R_{\text{EPE}}$ is distinct from the Clean Error (accuracy w.r.t. ground truth):
> $$R_{\text{EPE}}=\text{EPE}[f(I), f(I_c)], \quad \text{Clean Error}=\text{EPE}[f(I), g],$$
> where $g$ is the ground-truth flow. Hence, a small $R_{\text{EPE}}$ does not mean that the corrupted case became "easier" but that the model prediction remained stable between clean and corrupted inputs. *The robustness metric never rewards improved accuracy on corrupted data; it only quantifies stability.*
>
> To avoid confusion, we visually separated these two quantities in Tab. 1 by adding a double line and a distinct symbol for Clean Error ($\varepsilon_{\text{clean}}$), and explicitly emphasize in Sec. 3.2 that lower robustness scores correspond to higher stability, not improved accuracy.
>
> We thank the reviewer again for these thoughtful questions and believe that the clarified formulations made the metric definitions and their interpretation substantially better in the revised paper.

---

### Official Review · Reviewer_6GQ8 · 2025-10-30

**Soundness:** 3
**Presentation:** 3
**Contribution:** 1
**Rating:** 4
**Confidence:** 3

**Summary:**

The paper updates a recently published benchmark for optical flow and scene flow (SPRING , Mehl et al 2023) so that it also includes image corruptions (e,g. blurring the image, simulating bad weather etc.). It also provides a metric for evaluating algorithms and provides an initial benchmark of existing algorithms.

**Strengths:**

Datasets help advance the field of computer vision and it is important to evaluate algorithms in more challenging settings (especially since the Spring dataset is based on computer graphics and there is always the danger that algorithms will overfit to the particulars of the renderer).

**Weaknesses:**

I am afraid that the contribution is too limited in my mind for a conference publication. The results beyond the introduction of the dataset are very limited and I did not see any insights that have been obtained so far from using this dataset. The contribution would be better appreciated in a "datasets and benchmarks" track in a major conference but as far as I know, ICLR does not have such a track.

**Questions:**

Can you formulate any insights you have learned from the current benchmarking?

---

> ### Author Response · Authors · 2025-11-27
>
> Thank you for your review, and for acknowledging the importance of robustness evaluation for advancing computer vision benchmarks.
> In the following, we would like to emphasize our contributions and demonstrate the insights we have already gained regarding robustness benchmarking.
>
> **Scope and contribution of the work.**
>
> While we used the Spring dataset as base for building a robustness benchmark, the contribution of RobustSpring go well beyond simple modifications of the Spring data. With RobustSpring we introduce
>
> - a corruption generation pipeline that integrates image degradations in time, stereo, and depth for dense correspondence tasks,
> - a ground-truth-free robustness metric based on Lipschitz continuity, that allows to systematically disentangle accuracy and robustness measurements, and
> - a benchmarking setup that is ready to use for the scientific community, with benchmarking website and several design considerations to make it scalable and lightweight.
>
> Together, these elements make RobustSpring the first benchmark to systematically measure and compare robustness for optical flow, scene flow, and stereo. To the authors' knowledge, this is a capability not provided by any prior dataset.
>
> **Insights gained from benchmarking.**
>
> Beyond establishing the benchmark, our experiments reveal several insights into robustness benchmarking, which we want to highlight here:
>
> - *Disentangling accuracy and robustness.* As shown in Fig. 4c, our experiments demonstrate that accuracy and robustness are only weakly correlated. Some corruption classes align the two, while others show divergent behavior. This confirms that robustness cannot be inferred from accuracy alone and should be treated as an independent evaluation axis, which is one of the core motivations for RobustSpring. We added a more detailed discussion of accuracy–robustness relations to Sec. 4.1, Optical Flow.
> - *Corruption-specific failure modes.* We show that specific corruptions (e.g., weather-based corruptions like rain and snow) consistently yield stronger degradations across all methods, indicating that they are underrepresented in current training pipelines or not being handled well by current model architectures. This provides a concrete direction for future research.
> - *Metric ranking.* Our comparison of Average, Median, and Schulze rankings shows that different aggregation metrics lead to different model orderings. This shows that benchmark designers and users must choose summarization methods carefully. RobustSpring makes these differences explicit.
> - *Benchmark utility for method analysis.* In Sec. 4.1, we include a brief architectural comparison that illustrates how RobustSpring can surface characteristic robustness behaviors across model families. For example, transformer-based architectures (e.g. GMFlow, FlowFormer) achieve high overall robustness but degrade strongly under pixel-level noise, whereas hierarchical convolutional models (e.g. MS-RAFT+) maintain more balanced robustness across corruptions. Stacked architectures like FlowNet2, though less accurate, show surprising resilience to noise. These findings demonstrate that architectural design affects robustness in systematic and interpretable ways (insights that were previously unavailable without a dedicated corruption benchmark) and serve as an example of the type of analysis users can perform when evaluating or developing new methods.
>
> **Fit to ICLR.**
>
> In its call for papers, ICLR explicitly welcomes benchmarks that enable the systematic evaluation of learned systems.
> RobustSpring directly targets the robustness of deep learning models for dense prediction, providing a tool to study model generalization and reliability, which are key considerations when evaluating the usefulness of learned representations for computer vision, which is also listed in the ICLR call for papers.

---

> > ### Comment · Reviewer_6GQ8 · 2025-11-28
> > **Thank you for the response**
> >
> > Thank you for your response. I will keep my original score. As I wrote in my original review, I would not mind if the paper is accepted but in my mind the contribution is too limited.

---

### Official Review · Reviewer_gjCL · 2025-10-31

**Soundness:** 2
**Presentation:** 3
**Contribution:** 3
**Rating:** 4
**Confidence:** 5

**Summary:**

This paper introduces RobustSpring, a large-scale benchmark designed to systematically evaluate the robustness of optical flow, scene flow, and stereo models against 20 types of image corruptions, such as blur, noise, compression artifacts, and weather effects. The benchmark extends the Spring dataset by applying corruptions that are consistent in time, stereo, and depth. A new corruption robustness metric based on Lipschitz continuity is proposed to quantify model stability without relying on ground truth. Experiments across 16 models demonstrate large robustness variations across corruption types, and analyses show that robustness partially correlates with accuracy and transfers to real-world data.

**Strengths:**

(1) Comprehensive Benchmark Design. The paper introduces the first unified benchmark for evaluating robustness across optical flow, scene flow, and stereo tasks, with a carefully designed corruption taxonomy and consistency mechanisms.

(2) Principled Metric and Evaluation Protocol. The use of a ground-truth-free Lipschitz-based robustness metric provides a theoretically sound and practical approach to isolate robustness from accuracy, avoiding common confounds in prior evaluations.

(3) Extensive and Systematic Experimental Validation. The authors benchmark a wide range of models, analyze architecture-specific trends, and validate the transferability of robustness metrics to real-world datasets (e.g., KITTI), reinforcing the benchmark’s practical relevance.

**Weaknesses:**

(1) Limited Analysis Beyond Benchmarking. While the benchmark is well-motivated, the paper mainly presents benchmark results without deeper analysis of why certain architectures or design choices improve robustness (e.g., effect of global vs. hierarchical feature aggregation).

(2) Single Severity Level per Corruption. Only one corruption severity level is used to reduce computational cost, which limits fine-grained analysis of model sensitivity. A scalability study across multiple severities could provide more insight into robustness trends.

(3) Lack of Ablation or Analysis Experiments. The work could benefit from additional analysis experiments, such as varying the SSIM threshold used for corruption tuning, testing the effect of different subsampling ratios, or studying the sensitivity of the Lipschitz-based robustness metric’s formulation.

(4) Limited Discussion of Generalization and Scope. The paper acknowledges only briefly that the benchmark does not cover the full corruption space. A clearer discussion of future extensions (e.g., dynamic lighting, motion artifacts beyond the current set) or domain-specific customization would strengthen its impact.

**Questions:**

See the weakness.

---

> ### Author Response · Authors · 2025-11-27
>
> Thank you for your constructive feedback. We appreciate the positive assessment of our benchmark design, theoretical formulation, and extensive experimental validation. Below, we address each of the concerns in detail.
>
> **(1) Analysis Beyond Benchmarking**
>
> This paper focuses on ablating the benchmark character of RobustSpring. We demonstrate this through multiple ablations showing
> (i) the value of different ranking systems;
> (ii) the influence of data subsampling strategies;
> (iii) the validity of the robustness metric for object-centric corruptions; and
> (iv) the transferability of robustness results to real-world data.
> Regarding actionable insights into method development, our initial benchmark results reveal distinct robustness signatures across model families.
> Transformers demonstrate high overall robustness but are sensitive to pixel-level noise.
> Hierarchical models (e.g., MS-RAFT+) exhibit balanced robustness due to multi-scale aggregation.
> Stacked architectures (e.g., FlowNet2) are resilient to high-frequency distortions.
> RobustSpring, backed by the ablations above, provides the foundation for deeper analyses of how architectural designs and training protocols affect robustness. This new type of analysis allows users to evaluate or develop their methods.
>
> **(2) Severity Levels per Corruption**
>
> We acknowledge that a single severity level limits the granularity of robustness evaluation.
> Our choice was driven by practicality, since evaluating a single model across all 20 corruption types already produces over 2 TB of raw data per task.
> Increasing the number of severities would make public benchmarking computationally infeasible for most research groups.
> This design follows the precedent of other large-scale robustness benchmarks.
>
> **(3) Ablation or Analysis Experiments**
>
> We appreciate the reviewer's suggestions. Many of the proposed ablations were considered during the design of RobustSpring, and we want to discuss each of them below.
>
> *Varying SSIM thresholds*
>
> Although varying the SSIM target may seem natural, SSIM sensitivity differs significantly across corruption types (e.g., noise is much more sensitive to SSIM than blur for comparable perceptual severity increase). A linear sweep of SSIM therefore would not yield a meaningful ablation, as different corruptions would change in perceptual strength at different rates, making relative robustness rankings difficult to interpret. To compare different SSIM thresholds fairly, one would need to ensure that all thresholds produce visually comparable corruptions across all methods, which requires a perception study for each threshold.
> We conducted a user study to validate the choice of corruption strengths and their associated SSIM values in RobustSpring.
> Please refer to App. A.6 for more details.
>
> *Different subsampling ratios*
>
> We investigate different subsampling ratios in Sec. 4.2 and Tab. 2a. They show that the results on 0.05% subsampling closely match the ones on 1% and the full evaluation, confirming that our method is stable while being computationally efficient.
>
> *Sensitivity of the metric*
>
> Would you be able to give more details on the type of sensitivity analysis that you would like to see for the robustness metric?
> Because corruptions are SSIM-equalized, the input perturbation magnitudes are comparable across all corruption types, making normalized and unnormalized variants effectively equivalent up to a constant scaling. Additional formulations would therefore not alter the ranking or interpretation of robustness, but increase the computational load.
>
> **(4) Discussion of Generalization and Scope**
>
> Our corruption set builds on the five perturbation families of Hendrycks & Dietterich (2019), which capture the most common degradation sources in real-world imagery (weather, color, blur, noise, and camera artifacts).
> The selection was inspired by their 15 corruptions but adapted for dense matching to ensure consistent, efficient, and, sometimes more realistic implementations (Sec. 3.1).
> We additionally include five practically relevant corruptions: saturation, Gaussian blur, speckle noise, rain, and spatter.
> The blur and noise categories cover the most frequent camera and sensor perturbations. Weather models major outdoor disturbances but currently omits strong lighting effects and detailed scene-dependent illumination.
> Quality degradations focus on per-frame artifacts such as JPEG compression and do not include temporal video-compression effects.
> Color corruptions are limited to basic post-processing changes; more realistic variations (e.g., colored or dynamic illumination) would require re-rendering scenes in Blender rather than post-hoc adjustments.
> Promising extensions include:
>
> - Weather: bloom, glare, dusty conditions.
> - Color: gamma changes, low-light scenarios, colored or dynamic (space- or time-varying) illumination.
> - Artifcats: JPEG 2000, additional video-codec distortions.
>
> We added a concise version of this in Sec. 3.1.

---

### Author Response · Authors · 2025-12-02
**Summary of Discussion and Revisions**

We would like to thank the reviewers for their detailed feedback, which helped us improve both the scope and the technical clarity of the paper. They consistently highlighted RobustSpring as the first unified benchmark for robustness in optical flow, scene flow, and stereo (gjCL), emphasized the importance of evaluating models under challenging real-world degradations (6GQ8), praised its large and diverse corruption design with principled metrics and efficient evaluation (qH6X), and noted that no prior benchmark assesses robustness to image corruptions for dense correspondence while commending the decoupling of accuracy and robustness (oJUR).

Below we summarize the main changes and additions made in response to the reviews:

- **Additional experiments:** Included SEA-RAFT (ECCV 2024), validated corruption strengths (SSIM) in a perceptual user study (App. A.6), and discussed ablations on subsampling ratios (Sec. 4.2 and Tab. 3a).
- **Deeper analysis:** Expanded study of accuracy vs. robustness (updated Fig. 4c and Sec. 4.1), clarified corruption-specific failure modes, and highlighted architecture-dependent robustness behaviors.
- **Clearer corruption design:** Refined explanations of time/stereo/depth consistency (Sec. 3.1), justified corruption families by connecting each corruption family to common real-world degradation sources (Sec. 3.1), and explained the use of estimated depth to avoid ground truth leakage.
- **Clarifications:** Added precise definitions of norms and accuracy metrics (Sec. 3.2), clarified the Lipschitz-based metric formulation, explained relative robustness in the KITTI experiment, and fixed ambiguous phrasing.

These revisions strengthen the benchmark’s clarity and demonstrate that RobustSpring provides both a substantive robustness evaluation suite and actionable insights for understanding and improving learned dense matching models.

---

### Meta-Review · Area_Chair_WdBy · 2026-01-02

**Summary:**

This paper proposes a benchmark to evaluate robustness of optical flow, scene flow, and stereo models to 20 image corruptions ranging from blur, noise, quality to weather, extending with time-, stereo-, and depth-consistent corruptions.

**Reviewer Concerns:**

Most concerns were addressed:
- Limited insights beyond benchmarking: Authors clarified benchmark-focused ablations and articulated insights already observed, including weak correlation between accuracy and robustness, corruption-specific failure modes, and architecture-specific robustness.
- Metric clarity: Questions on “value difference,” relative robustness, and interpretation of robustness vs accuracy were clearly addressed
- Missing baselines / subsampling: SEA-RAFT was added and 0.05% subsampling shown to match full evaluation.

Concerns left unaddressed:
- Contribution scope / fit: One reviewer maintained the view that the contribution is too limited proposing it be a better fit to a benchmark track.
- Single severity level: Acknowledged as a limitation, justified by practical scalability.

**Reviewer Scores:**

gjCL (4)  likely to be 6, given rebuttal addressed raised concerns.
6GQ8 (4) remains 4, unchanged after rebuttal.
qH6X (6) likely 7, as clarification questions were resolved.
oJUR (8) remains 8, with concerns largely addressed.

Based on these I recommend accept as a poster

---

### Decision · Program_Chairs · 2026-01-26

Accept (Poster)